



# Technique for comparison of backscatter coefficients derived from in-situ cloud probe measurements with concurrent airborne Lidar

Shawn W. Wagner[1], David J. Delene[1]

[1]Atmospheric Sciences, University of North Dakota, Grand Forks, 58202, United States of America

*Correspondence to*: Shawn W. Wagner (shawn.wagner.atmos@gmail.com)

**Abstract**

Jet engine power loss due to ice particle accumulation is a recognized aviation hazard occurring in cloud conditions difficult to forecast or visually recognize. High-altitude cirrus clouds can have ice particle concentrations high enough to be dangerous; therefore, pilots must be informed when aircraft enter such environments. One approach to determining ice

particle concentration is an onboard Lidar system. Concurrent Lidar measurements are compared to backscatter coefficients derived from particle size distributions obtained from wing-mounted, in-situ probes during four case studies consisting of sixty-second flight segments at different temperatures; +7 °C and +4 °C for water droplet analysis, -33 °C and -46 °C for ice particle analysis. Backscatter coefficients derived from external cloud probes (ECP) are correlated (0.91) with measurements by an airborne Lidar system known as the Optical Ice Detector (OID). Differences between OID and ECP backscatter

coefficients range from less than 1 to over 3 standard deviations uncertainties. The backscatter coefficients are primarily in agreement for liquid clouds and disagreement for the -33 °C and -46 °C cases, with ECP derived backscatter coefficients lower than the OID for three out of the four cases. Measurements over four research flights show that measured total water content is correlated (0.74) with the OID backscatter coefficient, which indicates that the OID is a useful instrument for determining ice particle concentrations over a broad range of environments, including at ice water contents as low as 0.02 g

m$^{-3}$. Additionally, concurrent measurements from cloud imaging probes and an airborne Lidar test system provides improved knowledge of cloud conditions and can help in understanding cloud processes.

## 1 Introduction

Airborne ice particle ingestion into the engines of high-altitude jets are a serious aviation hazard in certain conditions. A number of power loss events caused by ice ingestion have occurred since 1990 (Lawson et al., 1998), which prompted the

Federal Aviation Administration to issue Airworthiness Directive 2013-NM-209-AD (Airworthiness Directives; The Boeing Company Airplanes, 2020). 2013-NM-209-AD requires flight manuals of Boeing models 747-8, 747-8f, and 787-8 to advise the attending flight crew of potential ice particle icing conditions at high altitudes, and prohibits flight operation in those



conditions. Such regulations create the need for instruments that alert flight crews whenever the aircraft enter dangerous environments, which can include sub-visible and radar undetected, ice clouds. The Optical Ice Detector (OID) is an onboard,

short-range cloud Lidar that is under development by Collins Aerospace to detect and characterize hazardous cloud environments (Ray et al., 2009; Halama et al., 2010; Ray and Anderson, 2015). As with any cloud Lidar, the OID measures backscatter, which depends mainly on the number and size of liquid droplets and ice crystals.

The objective is to evaluate the 2015 OID instrument using backscatter coefficient measurements and backscatter coefficients derived from particle size distributions obtained from state-of-the-art cloud probe observations. The evaluation

uses backscatter coefficient uncertainty obtained by summing each size channel's uncertainty, which is derived using both particle concentration and channel width. Quantitative comparisons of OID backscatter to research grade cloud probes is a major step in the deployment process of a commercial, onboard Lidar system to alert flight crews of dangerous ice concentration conditions. Additionally, such comparisons enable evaluation of backscatter theory, the processing of measurements, and the interpretation of observations.

## 2 Background

Since the early 1990s, there have been over 240 ice ingestion-related incidents involving commuter and large transport aircraft (Mason et al., 2006). Many of these aircraft incidents occurred at altitudes greater than 3000 m above MSL (mean sea level) (Bravin and Strapp, 2019) and near convective clouds (Haggerty et al., 2018). Before the early 2000s, ice particles in convective clouds were believed not to adhere to cold engine components and hence were not considered a safety hazard.

However, power loss incident frequency continued to increase as the number of high-altitude flights increased. Analysis of 46 power loss events above 3000 m MSL (Mason et al., 2006) indicated that aircraft would gradually lose power, with some aircraft experiencing total engine shut down. Once the aircraft descended to below 3000 m MSL, most failed engines were restarted and normal engine performance was restored. However, in some cases engine damage was sustained.

In 2002, a transport aircraft with dual Rosemount Icing Detectors experienced engine power loss without the presence of

supercooled liquid water (Mason et al., 2006). Post event analysis concluded that ice ingestion-related shutdowns can take place in environments consisting entirely of ice crystals, not simply in environments with super-cooled liquid water. This aircraft hazard from ice particles causing engine power loss was termed "ice particle icing" (Mason et al., 2006) or "ice crystal icing" (Haggerty et al., 2018) to differentiate it from "icing" of surfaces from the impaction and accretion of super-cooled liquid droplets. In 2013, an event occurred above 10 km that caused permanent damage to engine compressors and

prompted an Airworthiness Directive issuance (Airworthiness Directives; The Boeing Company Airplanes, 2020) on 27 November 2013.



Cirrus clouds can contain ice crystal concentrations high enough to impact the performance of aircraft engines (Gayet et al., 2012; Heymsfield, 1986) and pitot tubes (Jarvinen, 2013). Analysis of 22 mid-latitude cirrus clouds shown an average ice particle concentration of 1 cm$^{-3}$, with some concentrations above 5 cm$^{-3}$ (Lawson et al., 2006a). However, most cirrus clouds

have concentrations from 0.01 cm$^{-3}$ to 1 cm$^{-3}$ (Krämer et al., 2009). Typically, ice crystal concentration increases with higher updraft speed and the subsequent increase in supersaturation (Heymsfield and Miloshevich, 1993). Frey et al. (2011) found that outflow from developing mesoscale convective systems in Africa contained ice particle concentrations as high as 8.3 cm$^{-3}$, with sub-visible cirrus having an average ice particle concentration of 0.01 cm$^{-3}$. See Heymsfield et al. (2017) for additional information on cirrus cloud properties and processes.

Due to their potentially high concentrations, and often lack of visibility to pilots and on-board radar, cirrus clouds pose an aircraft flight safety hazard due to potential engine power loss. Mitigating this hazard requires alerting flight crews of dangerous ice particle concentrations using onboard instrumentation. Research grade aircraft instruments for measuring ice particle concentration have been available since the 1970s (Baumgardner et al., 2017). Optical array probes are a class of in-situ instruments that measure individual particle size using the blockage of laser light (shadows) on an array of diodes (Fig.

1). The fundamental measurement is the two-dimensional shadow area of individual particles, which are generally used to obtain size distributions by deriving the particle diameter. However, the fundamental measurement of a cloud imaging probe is particle shadow area where light is decreased by 50 % or more.





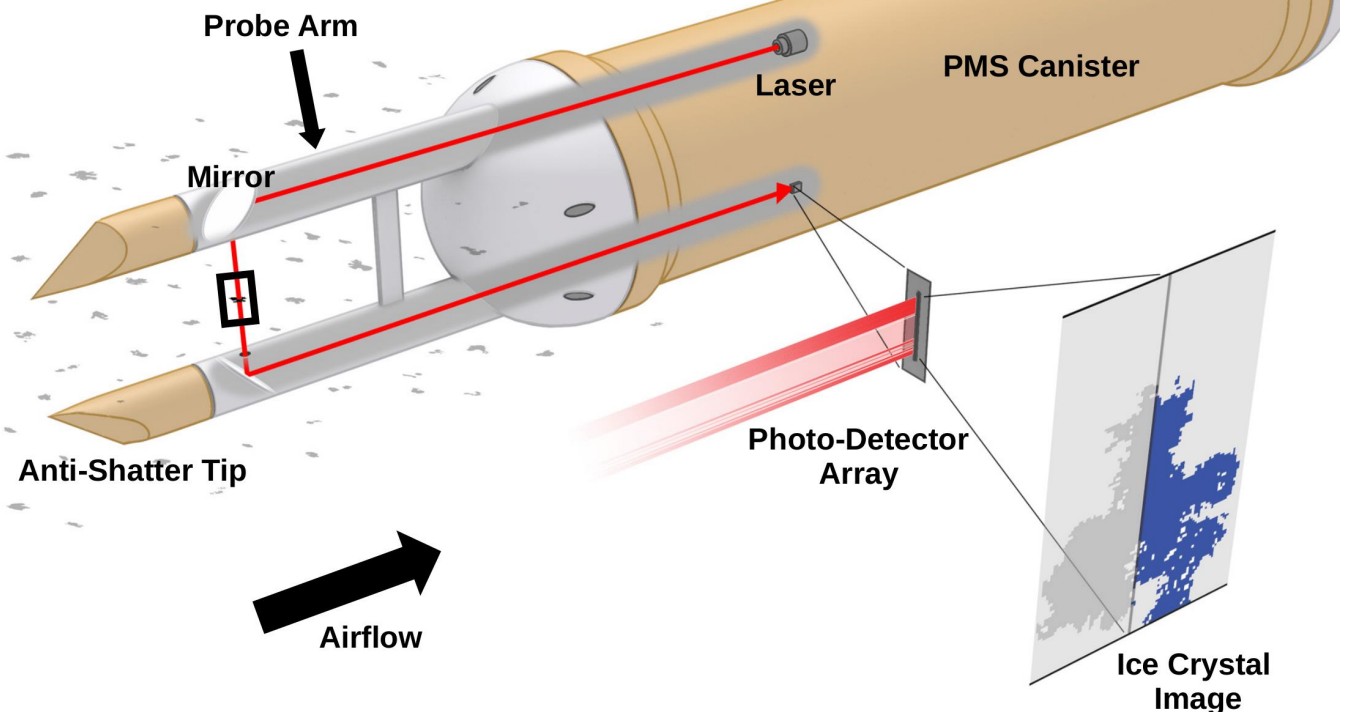


**Figure 1: Illustration showing the side view of a two-dimensional optical array probe in a Particle Measurement Systems, Inc., (PMS) canister. Images are obtained as particles pass through the laser beam (solid, red line) when the aircraft's speed creates airflow past the probe. The laser beam is directed between the probe arms using two 90 degree mirrors. Particles passing through the laser beam block light causing reduced illumination on some diodes in the photo-diode array. Images are produced when at**
**least one array element is reduced in intensity by a set amount (e.g., > 50 %). The depth of field (the location where the particle must cross the beam to be clearly imaged) is indicated with a black rectangle. The sampling frequency of the photo-detector array is adjusted using the measured true air speed, so images (example given in lower right) have symmetric pixel elements. Heated, anti-shattering tips prevent ice accretion and reduce the number of shattered particles that enter the sample volume (Korolev and Isaac 2005). Credit: Nowatzki, 2019.**

To avoid placing probes directly into the air stream, instruments have been developed that use flush-mounted windows which allow observations without altering the air flow around an aircraft. One such system is the Droplet Measurement Technologies (DMT) Backscatter Cloud Probe (BCP), which measures the backwards scattering of a continuous wave laser beam from cloud water droplets and ice crystals outside the aircraft (Beswick et al., 2014). The BCP has a sample volume of approximately 125 cm$^3$ s$^{-1}$ (at an aircraft speed of 100 m s$^{-1}$) that is located approximately 4 cm from the aircraft skin. The
BCP has been deployed on commercial aircraft to measure the particle size distribution (Beswick et al., 2015). BCP measurements have been evaluated (Beswick et al., 2015) using comparisons with a Cloud Droplet Probe (CDP) and a Cloud and Aerosol Spectrometer (CAS) probe.





The OID is similar to the BCP in using an observation window; however, the OID uses a Lidar with a conical sample volume that extends up to 10 m from the aircraft (Ray and Anderson, 2015). The OID sample volume is 4500 cm$^3$ s$^{-1}$ at 100 m s$^{-1}$ with a viewing direction perpendicular to the forward motion of the aircraft (Fig. 2), and uses a circularly polarized wavelength of 905 nm and a randomly orientated, linearly polarized wavelength of 1550 nm. The 905 nm beam enables measurement of the fourth Stokes parameter (V) (Liou and Yang, 2016; Hulst, 1981) and is the focus of the study, while the 1550 nm wavelength channel is not used. The OID's large sample volume, ability to observe in the unperturbed air ahead of the aircraft, and flush mounting are important for commercial applications in detecting hazardous aircraft environments. Additionally, the OID's larger sample volume compared to other aircraft instruments is advantageous for comparison with other remote sensing platforms (e.g., CALIPSO). Hence, a research grade OID type instrument would be an important tool for linking small scale microphysical cloud observations to large scale remote sensing observations.

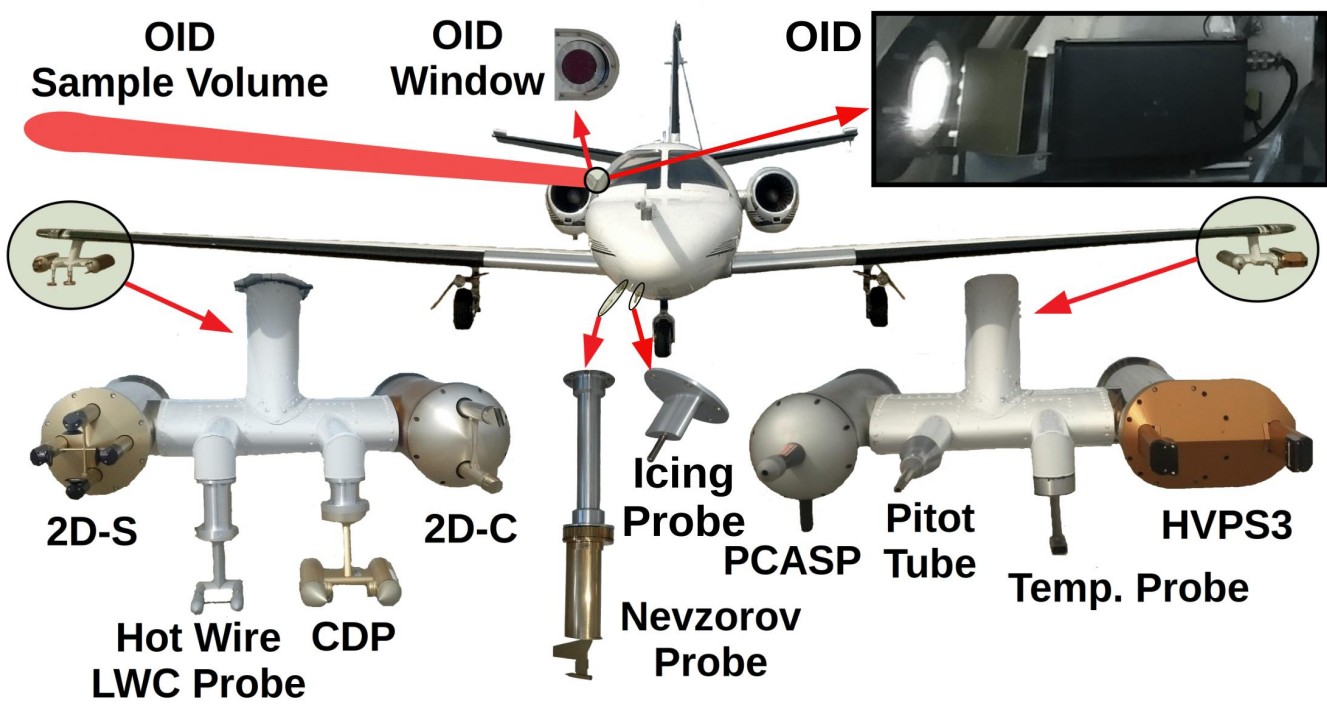

**Figure 2:** Image showing the North Dakota Citation Research Aircraft with an enlarged view of the wing-tip pylons, the Nevzorov probe and the Rosemount Icing Probe as mounted for the 2015 field project over Cape Canaveral, Florida (CAPE2015). OID Window (upper center) is the viewing port through the pressurized cabin for the Optical Ice Detector (OID). The OID is angled slightly (10 to 15 degrees) forward of the wing so the sampling region (OID Sample Volume) is in front of and above the right wing. An image of the OID in the Citation fuselage is shown in the upper right. The Cloud Droplet Probe (CDP) measures particles from 2 to 50 μm (30 channels) using forward light scattering. The Two-Dimensional Stereo (2D-S) probe measures particles using shadowed diodes (two (one horizontal and one vertical) 128 element arrays) of 10 μm. The High-Volume Precipitation Spectrometer Version 3 (HVPS3) measures particles using shadow diodes (128 element array) of 150 μm. The Two Dimensional Cloud probe (2D-C) is an older probe that also measures particles using shadowed diodes (32 element array) of 30 μm. The Rosemount Icing Probe detects super-cooled liquid water using a vibrating metal rod. The Nevzorov probe is a hot-wire





**probe that measures liquid and total water content (TWC). Subtracting the liquid water content LWC from the TWC determines the ice water content (IWC). The King Hot Wire Liquid Water Content Probe is an older probe that only measures LWC. The Passive Cavity Aerosol Spectrometer Probe (PCASP) model measures aerosols from 0.1 to 3.0 μm. The Rosemount temperature probe (Temp. Probe) measures ambient air temperature, and the pitot tube measures air speed using a differential pressure transducer connected to a static and a forward facing port.**

## 3 Measurements

The OID has been deployed on the North Dakota Citation Research Aircraft (Delene et al., 2019) during several field projects, including a 2015 field project (CAPE2015) to study Florida thunderstorms (Schmidt et al., 2019). The Citation Research Aircraft has conducted several research projects to collect cloud microphysical observations using various instrumentation configurations (Skofronick-Jackson et al., 2014; Jensen et al., 2015; Delene, 2016). Multiple field projects have included OID measurements; however, this study analyzes data (Wagner and Delene, 2020a) only from CAPE2015

flights, which focused on measurements of cirrus cloud anvils from convective storms. An OID is mounted in the Citation Research Aircraft's fuselage to measure backscattered light from cloud particles slightly ahead of and along the span of the aircraft wing (Fig. 2). The OID's viewing port is fitted with anti-reflection coated class. A fan moves cabin air across the port window to prevent water and ice condensation on the inside surface. Interested readers are referred to Ray and Anderson (2015) for a thorough description of the OID instrument and how it obtains measurements, as only a brief summary is

provided herein.

The OID measurement of backscatter coefficient (β in units of km$^{-1}$ ster$^{-1}$) assumes a homogeneous cloud particle distribution over the sampling distance R. The backscatter coefficient is calculated by inverting the returned Lidar power P(R) equation:

$$P_R = \beta G_R e^{-2\varepsilon R}, \tag{1}$$

where G(R) is a light collection efficiency as a function of particle range, ε is the extinction coefficient, and R is the range to the particles (Ray and Anderson, 2015). The OID emits light pulses at a repetition rate of 20 kHz, with each pulse having a temporal width of 4 ns full width at half maximum (Halama et al., 2010). The 20 kHz measurements are aggregated to produce 5 Hz raw data. Raw data is averaged to match the 1 Hz external cloud probe (ECP) processed data. The 1 Hz frequency data has more uncertainty than 5 or 10 s averaged data; however, 1 Hz data is able to detect rapid changes in cloud

conditions that are important when monitoring environmental conditions from a fast-moving aircraft and for understanding cloud processes. While the OID transmits both 905 nm and 1550 nm laser light, absorption at 905 nm by water is approximately 1000 times less than at 1550 nm. Hence, using only the 905 nm wavelength simplifies analysis by eliminating the consideration of absorption. While a complete OID error analysis is not yet available, the primary error source is likely the inversion of the range-resolved Lidar signal to estimate extinction. For additional details regarding Lidar retrievals, see

Lolli et al. (2013).



The North Dakota Citation Research Aircraft is a Cessna model 550, fan-jet aircraft that has instruments to measure atmospheric state parameters such as temperature, relative humidity, and wind velocities, as well as the cloud particle size distribution using a set of in-situ cloud probes. In-situ cloud microphysical instrumentation (Fig. 2) includes a Stratton Park Engineering Company Incorporated (SPEC) Two-Dimensional Stereo (2D-S) probe (Lawson et al., 2006b), a SPEC High-

Volume Precipitation Spectrometer Version Three (HVPS3) probe (Kumjian et al., 2016), and a Sky Tech Research Incorporated Nevzorov probe (Korolev et al., 2013b). Optical array probes, such as the 2D-S and HVPS3, use a laser beam between probe arms directed onto an array of photo-diodes to observe shadowing by passing cloud particles during flight (Fig. 1). The 2D-S has 128 diodes with a 10 µm resolution and the HVPS3 has 128 diodes with a 150 µm resolution. Ice crystals and water droplets passing through the laser beam between the arms block light, causing reduced illumination on

some of the individual elements of the photo-diode array. Images are produced when at least one array element is "shadowed" (i.e. reduced in intensity by 50 % or more). The sampling frequency of the photo-detector array is adjusted using the measured true air speed to produce correctly scaled cloud particle shadow images produced by concatenating successive scans of the diode array. When cloud particle concentrations are high, the 2D-S probe may not have sufficient time to off-load the array buffers as quickly as they are filled, which results in reduced probe activity (Lawson et al., 2006b).

CAPE2015 has no "dead time" issues as the particle concentration is low enough that the probe's activity is 100 % during all flights analyzed.

The Nevzorov probe is a hot-wire instrument with sensors to measure cloud liquid and total (liquid + ice) water content (Korolev et al., 1998). The total water content (TWC) sensor uses a conical receiver to collect both liquid water droplets and ice crystals, while the liquid water content (LWC) sensor uses a round wire to collect droplets but very few ice particles.

Both sensors have corresponding reference sensors that are exposed to the airflow but not cloud particles. The additional power required to maintain constant temperature for particle sensors compared to reference sensors is directly related to water mass. Mass measurements and aircraft true airspeed are used to determine LWC and TWC. Ice water content (IWC) is not directly measured but calculated by subtracting LWC from TWC. Nevzorov probe data processing uses multiple linear regressions of the measured static pressure and indicated airspeed to obtain a per-flight calibration (Schwarzenboeck et al.,

2009), and an automatic baseline correction ensures out-of-cloud water content is zero (Delene et al., 2019). The uncertainty of Nevzorov measurements increases with hydrometeor size because large ice crystals may bounce out of the conical receiver before complete evaporation occurs, while large water droplets may not evaporate completely before bouncing from the conical receiver or shedding from the round wire. While it is possible to use the OID measurement to derive IWC (Anderson and Ray, 2019), the 2015 data set does not have a direct measurement of IWC. Hence, an IWC comparison would

have more uncertainty than a backscatter coefficient comparison.





In recent years, efforts have been made to collectively document the uncertainties associated with optical cloud microphysical probes (Baumgardner et al., 2017). These uncertainties include the possibility of particles counted in the wrong size channel (Korolev et al., 1991; Baumgardner and Korolev, 1997), sizing uncertainty for ice particles outside of the focal volume (Connolly et al., 2007), approximation of irregular ice particles as spheres (Wu and McFarquhar, 2016), and

particle coincidence (Cooper, 1988; Lance, 2012; Johnson et al., 2014). These uncertainties are accounted for as much as possible in the data processing and analysis methodology used here. Additionally, splashing of water droplets and the shattering of ice crystals can cause measurement errors; hence, anti-shattering probe tips and data processing methods have been employed to minimize such issues.

## 4 Data Processing

The 2D-S and HVPS3 images are processed to obtain particle concentrations using the sample volume (SV) given by:

$$SV = DOF * w * TAS * t, \tag{2}$$

where DOF is the depth of field, w is the effective width of the photo-diode array, TAS is the aircraft's true airspeed measured by a pitot tube, and t is elapsed time (McFarquhar et al., 2017). The depth of field (Fig. 1) is the region along the laser beam where particles are sufficiently within focus to create a clear shadow on a photo-diode array (Korolev, 2007).

DOF*w is the sample area, while TAS*t provides the third dimension in determining SV. The nose gust boom measured TAS and left wing pitot tube (Fig. 2) measured TAS agree within 3 % for the CAPE2015 field project.

Quality control conducted during the CAPE2015 field project involved instrument maintenance and performance checks. Instrument maintenance included cleaning the outside windows on all optical probes before each flight. Performance checks included reviewing the voltages of the first and last (edge) photo-diodes in the cloud probe arrays and ground testing

instruments with spray water to ensure correct performance. To enable timely post-flight data review, the open source Airborne Data Processing and Analysis (ADPAA) software package (Delene, 2011) is used to automatically process and visualize data after each flight. Even with rigorous instrument quality control and assurance, data set problems can still occur. For example, the liquid cloud droplet size spectrum measured by a DMT CDP is inconsistent (i.e. a factor of ten low) compared to the 2D-S size spectrum in the overlapping region. Misalignment of the CDP laser discovered after the

CAPE2015 project makes the CDP measurements validity questionable; therefore, only the 2D-S and HVPS3 instruments are used to create a combined size spectrum.

All CAPE2015 two-dimensional optical array probe data are automatically processed using ADPAA code (Delene et al., 2020) that interfaces with the System for Optical Array Probe Data Analysis Version 2 (SODA2) software package (Bansemer, 2013). Even when using Korolev anti-shatter tips, bursts of particles over a short sampling period are likely



shattering artifacts (Field et al., 2006); therefore, SODA2's shattering artifact rejection methodology is used for all CAPE2015 data processing. SODA2 processing of the CAPE2015 2D-S and HVPS3 images uses the center-of-mass-in method (Heymsfield and Parrish, 1978) to obtain particle concentrations. An ADPAA script merges 2D-S horizontal arms size channels (5 to 1000 µm) and HVPS3 (1000 to 30 000 µm) size channels obtained with SODA2 processing to create a combined particle size distribution ranging from 5 to 30 000 µm, which is used for the analysis (Table 1). The 2D-S

vertically oriented arm measurements are not used in the analysis; however, the two orientations have similar particle size distributions (see supplemental material). The CAPE2015 data has been processed and analyzed (Wagner, 2020) using the full reconstruction method (Heymsfield and Parrish, 1978); however, the center-of-mass-in method is used here since HVPS3 measurements are available to cover the larger particle size range, and the observed cirrus clouds do not have particles large enough to be excluded by center-of-mass-in HVPS3 processing.






**Table 1. Table listing parameters for the particle size distribution created by combining Two-Dimensional Stereo (2D-S) and High-Volume Precipitation Spectrometer Version 3 (HVPS3) probe measurements. The Number Column lists the combined spectrum channel number, the Probe Column indicates which probe made the measurements, and Bin Columns gives the original probe's channel number and total channels. The Size Range Column gives the lower and upper size of particles within the channel. The**
**Size Parameter Column gives the scattering size parameter (Eq. (3)) range for the lower and upper particle sizes and a wavelength of 905 nm. The Water Column lists backscatter efficiency for water particles, and the Ice Column lists backscatter efficiency for ice particles. The Q Ratio Column gives ratio of backscatter efficiency of water to the backscatter efficiency of ice. The MiePlot software package is used to calculate average efficiencies using particle diameters that span the combined spectrum width.**

| Number | Probe | Bin | Size Range | Size Parameter | Q(Water) | Q(Ice) | Q Ratio |
|---|---|---|---|---|---|---|---|
| 1 | 2D-S | 1 / 29 | 5 to 15 µm | 17 to 52 | 0.1184 | 0.1276 | 0.9276 |
| 2 | 2D-S | 2 / 29 | 15 to 25 µm | 52 to 87 | 0.1128 | 0.1204 | 0.9369 |
| 3 | 2D-S | 3 / 29 | 25 to 35 µm | 87 to 122 | 0.1137 | 0.1412 | 0.8052 |
| 4 | 2D-S | 4 / 29 | 35 to 45 µm | 122 to 156 | 0.1142 | 0.1463 | 0.7806 |
| 5 | 2D-S | 5 / 29 | 45 to 55 µm | 156 to 191 | 0.1183 | 0.1456 | 0.8125 |
| 6 | 2D-S | 6 / 29 | 55 to 65 µm | 191 to 226 | 0.1172 | 0.1398 | 0.8383 |
| 7 | 2D-S | 7 / 29 | 65 to 75 µm | 226 to 260 | 0.1280 | 0.1355 | 0.9446 |
| 8 | 2D-S | 8 / 29 | 75 to 85 µm | 260 to 295 | 0.1269 | 0.1328 | 0.9556 |
| 9 | 2D-S | 9 / 29 | 85 to 95 µm | 295 to 330 | 0.1401 | 0.1247 | 1.1235 |
| 10 | 2D-S | 10 / 29 | 95 to 105 µm | 330 to 364 | 0.1399 | 0.1184 | 1.1816 |
| 11 | 2D-S | 11 / 29 | 105 to 125 µm | 364 to 434 | 0.1546 | 0.1137 | 1.3597 |
| 12 | 2D-S | 12 / 29 | 125 to 145 µm | 434 to 503 | 0.1634 | 0.1032 | 1.5833 |
| 13 | 2D-S | 13 / 29 | 145 to 175 µm | 503 to 607 | 0.1809 | 0.0932 | 1.9410 |
| 14 | 2D-S | 14 / 29 | 175 to 225 µm | 607 to 781 | 0.2070 | 0.0786 | 2.6336 |
| 15 | 2D-S | 15 / 29 | 225 to 275 µm | 781 to 955 | 0.2277 | 0.0663 | 3.4344 |
| 16 | 2D-S | 16 / 29 | 275 to 325 µm | 955 to 1128 | 0.2547 | 0.0590 | 4.3169 |
| 17 | 2D-S | 17 / 29 | 325 to 400 µm | 1128 to 1389 | 0.2728 | 0.0508 | 5.3701 |
| 18 | 2D-S | 18 / 29 | 400 to 475 µm | 1389 to 1649 | 0.2966 | 0.0486 | 6.1029 |
| 19 | 2D-S | 19 / 29 | 475 to 550 µm | 1649 to 1909 | 0.3182 | 0.0424 | 7.5047 |
| 20 | 2D-S | 20 / 29 | 550 to 625 µm | 1909 to 2170 | 0.3434 | 0.0404 | 8.5000 |
| 21 | 2D-S | 21 / 29 | 625 to 700 µm | 2170 to 2430 | 0.3581 | 0.0369 | 9.7046 |
| 22 | 2D-S | 22 / 29 | 700 to 800 µm | 2430 to 2777 | 0.3860 | 0.0321 | 12.025 |
| 23 | 2D-S | 23 / 29 | 800 to 900 µm | 2777 to 3124 | 0.3988 | 0.0314 | 12.701 |
| 24 | 2D-S | 24 / 29 | 900 to 1,000 µm | 3124 to 3471 | 0.4157 | 0.0276 | 15.062 |
| 25 | HVPS3 | 5 / 28 | 1000 to 1200 µm | 3471 to 4166 | 0.4390 | 0.0270 | 16.259 |
| 26 | HVPS3 | 6 / 28 | 1200 to 1400 µm | 4166 to 4860 | 0.4793 | 0.0238 | 20.139 |
| 27 | HVPS3 | 7 / 28 | 1400 to 1600 µm | 4860 to 5554 | 0.5139 | 0.0238 | 21.592 |
| 28 | HVPS3 | 8 / 28 | 1600 to 1800 µm | 5554 to 6248 | 0.5532 | 0.0215 | 25.730 |
| 29 | HVPS3 | 9 / 28 | 1800 to 2200 µm | 6248 to 7637 | 0.5831 | 0.0190 | 30.690 |
| 30 | HVPS3 | 10 / 28 | 2200 to 2600 µm | 7637 to 9025 | 0.6419 | 0.0171 | 37.538 |


| 31 | HVPS3 | 11 / 28 | 2600 to 3000 µm | 9025 to 10 414 | 0.6952 | 0.0154 | 45.143 |
| 32 | HVPS3 | 12 / 28 | 3000 to 3400 µm | 10 414 to 11 803 | 0.7173 | 0.0147 | 48.796 |
| 33 | HVPS3 | 13 / 28 | 3400 to 3800 µm | 11 803 to 13 191 | 0.7722 | 0.0132 | 58.500 |
| 34 | HVPS3 | 14 / 28 | 3800 to 4200 µm | 13 191 to 14 580 | 0.8018 | 0.0130 | 61.677 |
| 35 | HVPS3 | 15 / 28 | 4200 to 4600 µm | 14 580 to 15 968 | 0.8290 | 0.0121 | 68.512 |
| 36 | HVPS3 | 16 / 28 | 4600 to 5000 µm | 15 968 to 17 357 | 0.8433 | 0.0122 | 69.123 |
| 37 | HVPS3 | 17 / 28 | 5000 to 6000 µm | 17 357 to 20 828 | 0.8479 | 0.0118 | 71.856 |
| 38 | HVPS3 | 18 / 28 | 6000 to 7000 µm | 20 828 to 24 300 | 0.8409 | 0.0116 | 72.491 |
| 39 | HVPS3 | 19 / 28 | 7000 to 8000 µm | 24 300 to 27 771 | 0.8017 | 0.0111 | 72.225 |
| 40 | HVPS3 | 20 / 28 | 8000 to 9000 µm | 27 771 to 31 242 | 0.7330 | 0.0110 | 66.636 |
| 41 | HVPS3 | 21 / 28 | 9000 to 10 000 µm | 31 242 to 34 714 | 0.6373 | 0.0104 | 61.279 |
| 42 | HVPS3 | 22 / 28 | 10 000 to 12 000 µm | 34 714 to 41 656 | 0.4916 | 0.0101 | 48.673 |
| 43 | HVPS3 | 23 / 28 | 12 000 to 14 000 µm | 41 656 to 48 599 | 0.3467 | 0.0100 | 34.670 |
| 44 | HVPS3 | 24 / 28 | 14 000 to 16 000 µm | 48 599 to 55 542 | 0.2962 | 0.0104 | 28.481 |
| 45 | HVPS3 | 25 / 28 | 16 000 to 18 000 µm | 55 542 to 62 485 | 0.3878 | 0.0098 | 39.571 |
| 46 | HVPS3 | 26 / 28 | 18 000 to 20 000 µm | 62 485 to 69 427 | 0.5822 | 0.0103 | 56.524 |
| 47 | HVPS3 | 27 / 28 | 20 000 to 25 000 µm | 69 427 to 86 784 | 0.9381 | 0.0098 | 95.725 |
| 48 | HVPS3 | 28 / 28 | 25 000 to 30 000 µm | 86 784 to 104 141 | 0.9826 | 0.0092 | 106.80 |

## 5 Methodology

Cloud particles are typically much larger than the wavelengths (905 nm and 1550 nm) of the OID, especially particles in cirrus cloud anvils. The ratio of particle circumference to wavelength defines the light scattering regime (geometric, Mie, or Rayleigh) and is known as the size parameter α, which is given by:

$$\alpha = \frac{\pi D}{\lambda},$$ (3)

where D is the cloud particle diameter, and λ is the wavelength of incident light (Hulst, 1981). Light scattering is in the

geometric regime when the size parameter is greater than 100, in the Mie regime when the size parameter is between 0.1 and 100, and in the Rayleigh regime when the size parameter is less than 0.1 (Bohren and Huffman, 1983). Most size channels of the combined cloud probe spectrum are in the geometric scattering regime (Table 1). However, determining backscatter coefficients from particle size distributions using geometric optics methods produces inaccuracies due to a lack of higher-order scattering terms (Yang and Liou, 1995; Zhou and Yang, 2015). While Mie theory strictly applies to spherical particles,

studies have found that uncertainties in using spherical Mie theory code for aspherical scatterers are far less than uncertainties associated with measurements of cloud particle sizes (Cairo et al., 2011). While geometric optics are often used for particles with a size parameter larger than 100 to conserve computational resources, Mie theory has no upper size limit and can be applied to the entire cloud probe size range.





Backscatter coefficients depend on particle scattering efficiency at 180 ° (backscatter efficiency), the number concentration,
and the particle size (Bohren and Huffman, 1983). The backscatter coefficient equation (Zhang et al., 2015 page 271) is
given by:

$$\beta_{ECP} = \sum_{i=1}^{i=max} Q_i \eta_i \pi r_i^2,$$ (4)

where βECP is derived backscatter coefficient from external cloud probes (ECP), i is particle size channel number (see Table
1), $n_i$ is number of particles in channel i, $Q_i$ is scattering efficiency at 180 ° for channel i, and $r_i$ is particle radius in channel i.
For water spheres, $\pi r^2$ is the cross-sectional area (A), while for irregular particles such as ice, A is modeled as the cross-
sectional area of an equivalent sphere. In either case, A is the particle area directly measured by the cloud imaging probe as
the area of shadowed diodes obtained using SODA2 data processing. Backscatter efficiency indicates effectiveness of a
particle at scattering in the 180 ° direction with regard to incident light. Backscatter efficiency is calculated using the Python
based software, MiePlot (Laven, 2018). Backscatter efficiency calculations use 20 ℃ for the medium temperature, 905 nm
for the incident light wavelength, and refractive indices of 1.3263 + 5.6 x $10^{-7}$j for water at 20 ℃ (Kedenburg et al., 2012)
and 1.3031 + 5.6 x $10^{-7}$j for ice (Warren and Brandt, 2008). The refractive indices depend not only on incident wavelength
but also on particle temperature, and therefore, density (Wesely, 1976). However, sensitivity tests using 0 ℃ and 10 ℃
water refractive indices show a difference of less than 1 % for particle diameters up to 105 µm. Since water at 4 ℃ has the
highest density (Tanaka et al., 2001), testing both 0 ℃ and 10 ℃ provides backscatter efficiency results over a range of
densities for water droplets. As ice particle density depends on multiple formation factors other than temperature, the effects
of ice particle temperature are less relevant and likely have little impact. Correct utilization of MiePlot for determination of
water backscatter efficiency has been confirmed by obtaining results comparable to Fig. 61 of Hulst (1981) and Fig. 1 of
Lolli et al. (2013).

Backscatter efficiency values are calculated using MiePlot for diameters distributed log-normally between 1 µm and 30 mm.
Backscatter efficiencies are averaged for all particle diameters within each channel (Shishko et al., 2020) in the combined
particle size spectrum (Table 1), beginning at 5 µm and ending at 30 mm (Fig. 3). Averaging over channel intervals smooths
out rapid changes in backscatter efficiencies with diameter. As a result, the backscatter efficiencies for water and ice are
nearly constant (within approximately 10 %) for particle diameters up to 105 µm (the upper limit of size bin 10 in Table 1).
Above 105 µm, water has a backscatter efficiency that increases, while ice has a backscatter efficiency that decreases. There
is a backscatter efficiency dip around 10 000 µm diameter, which is likely due to destructive interference of surface waves
(Hovenac and Lock, 1992). For more information about the effects of surface waves on light scattering, see Chýlek et al.
(1980). It should be noted that measurements from the OID directly involve backscatter and extinction coefficients, and thus
knowledge of the backscatter efficiency in relation to the OID is not necessary for this study.



**Figure 3: Plots showing backscatter efficiencies using Mie theory for spherical water (top) and ice (bottom) particles. The MiePlot software package (Laven 2018) is used to calculate efficiencies for 100 000 particle diameters distributed log-normally between 1 μm and 30 mm (red dots). Intervals range from 0.0001 μm for the smallest diameters and 3 μm for the largest diameters. Black circles are average efficiencies over the combined spectrum channel widths (see Table 1***Error! Reference source not found.* **for numerical values). A refractive index of 1.3263 + 5.6 x 10$^{-7}$j is used for water (Kedenburg et al. 2012) and 1.3031 + 5.6 x 10$^{-7}$j for ice (Warren and Brandt 2008). Scattering is for an air medium at 20 °C and incident light of 905 nm.**

Uncertainty in the backscatter coefficient ($\delta\beta$ECP) due to sizing and counting errors is derived using a weighted error propagation in quadrature method (Berendsen, 2011):





$$\delta\beta_{ECP} = \sum_{i=1}^{i=max} \sqrt{(Q_i \pi r_i^2 \delta\eta_i)^2 + (2\eta_i Q_i \pi r_i \delta r_i)^2}, \quad (5)$$

where $\delta n_i$ is concentration uncertainty for channel i and $\delta r_i$ is particle radius uncertainty for channel i. Poisson statistics determine absolute uncertainty in concentration:

$$\delta\eta_i = \eta_i * \max_{i=1}^{i=max} \left(\frac{1}{\sqrt{(N_i)}}\right), \quad (6)$$

where $N_i$ is counted particle number per channel (Horvath et al., 1990). Another example of applying Poisson statistics to cloud probe analysis can be found in Baumgardner et al. (2014). One half the channel width is used as the particle radius uncertainty, which is a lower bound on the uncertainty. Additional factors could result in a larger uncertainty. For example, particles outside the depth of field appear too large and hence not within the correct size channel (O'Shea et al., 2019).

The OID is undergoing development and is an immature instrument compared to two-dimensional array probes; hence, a full uncertainty analysis is not available and is beyond the scope of this paper. Instead, the 1Hz OID backscatter coefficient uncertainty (δβOID) is obtained from the standard deviation of the 5 Hz OID data. The standard deviation includes some natural variability, as the aircraft traverses different cloud conditions, in addition to random uncertainty; hence, the uncertainty is an upper bound. OID and ECP agreement is considered to occur when the measurement difference is less than 3 standard deviations (Berendsen, 2011), which is calculated using:

$$\sigma_3 = 3\sqrt{(\delta\beta_{OID})^2 + (\delta\beta_{ECP})^2}, \quad (7)$$

Thus, a time series plot of the OID and ECP backscatter coefficient difference is useful to determine periods of measurement agreement and disagreement. The third standard deviation is used as a threshold in accordance with the Three Sigma Rule (Pukelsheim, 1994) since the OID is in development and Eq. (7) does not include systematic errors (e.g. uncertainty in backscatter efficiency).

OID backscatter coefficients are compared to 1 Hz backscatter coefficients derived from the external cloud probes (2D-S, HVPS3) using a case study analysis of four, 60 s, CAPE2015 flight segments, with each segment selected to represent a different meteorological condition. The cases (Table 2) are at four different temperatures (+7 °C, +4 °C, -33 °C and -46 °C), which provide two warm cloud cases and two cold cloud cases at constant flight heading and altitude (Fig. 4). The warm cases (Fig. 4A and 4B) are at approximately 3 km altitude over the ocean, just off the east coast of Florida. The warm cloud cases have small liquid droplets that are not observable on radar (Table 2). The +7 °C case occurred early in the flight while waiting for air traffic control climb permission and sampled developing cumulus clouds with very few drizzle sized droplets. Similarly, the +4 °C case also occurred before ascending to sample cirrus clouds but sampled mature cumulus clouds with



drizzle sized droplets. As expected for these warm temperatures, the 2D-S images (Fig. 4A and 4B ) show small spherical droplets. The cold cases (Fig. 4C and 4D) occurred above 9 km in cirrus cloud anvils. Melbourne radar observations have cloud bases at 1000 m MSL (Table 2) for both cold cloud cases, which indicates sampling is above the stratus rain region, instead of the cirrus anvil outflow region of the storm. Cloud tops are located at 11 000 m MSL for the -33 °C case, and 15 000 m MSL for the -46 °C case. The 2D-S images show mostly irregularly shaped particles indicative of ice crystals. The -

33 °C case is warmer than the homogeneous freezing temperature (-38 °C) and has some small particles appearing round, which could be liquid, or frozen, cloud droplets. Therefore, the –33 °C case may contain liquid droplets; however, the -46 °C case, being much colder than the homogeneous freezing temperature, contains only ice particles.

**Table 2. Table listing details of the 60 s segments for the Florida, CAPE2015 case studies. Numbers in parentheses in the Flight Row indicate the first (1) or second (2) flight conducted on the given day. The cloud bases and cloud tops were derived from the**
**National Weather Service Melbourne radar. "Not Available" indicates insufficient radar reflectivity to detect the cloud. The TWC Row gives the average and standard deviation of the total water content (TWC) measured by the Nevzorov probe.**

| Case | +7 °C | +4 °C | -33 °C | -46 °C |
|---|---|---|---|---|
| Flight | 02 August 2015 | 01 August 2015 (1) | 31 July 2015 | 01 August 2015 (2) |
| Time | 69 510 to 69 570 sfm | 57 850 to 57 910 sfm | 71 710 to 71 770 sfm | 72 700 to 72 760 sfm |
| Altitude | 3043 m ± 17.2 m | 3440 m ± 3.3 m | 9479 m ± 3.8 m | 10 971 m ± 6.8 m |
| Temp | 6.7 °C ± 0.42 °C | 4.3 °C ± 0.11 °C | -33.0 °C ± 0.12 °C | -45.9 °C ± 0.27 °C |
| Latitude | 28.446° to 28.497° N | 28.725° to 28.761° N | 28.326° to 28.402° N | 28.408° to 28.480° N |
| Longitude | 80.4916° to 80.4859° W | 80.5915° to 80.5577° W | 80.445° to 80.4373° W | 80.714° to 80.686° W |
| Cloud Base | Not Available | Not Available | 1000 m | 1000 m |
| Cloud Top | Not Available | Not Available | 11 000 m | 15 000 m |
| TWC | 1.2 g m$^{-3}$ ± 0.47 g m$^{-3}$ | 0.5 g m$^{-3}$ ± 0.08 g m$^{-3}$ | 0.1 g m$^{-3}$ ± 0.05 g m$^{-3}$ | 0.1 g m$^{-3}$ ± 0.02 g m$^{-3}$ |



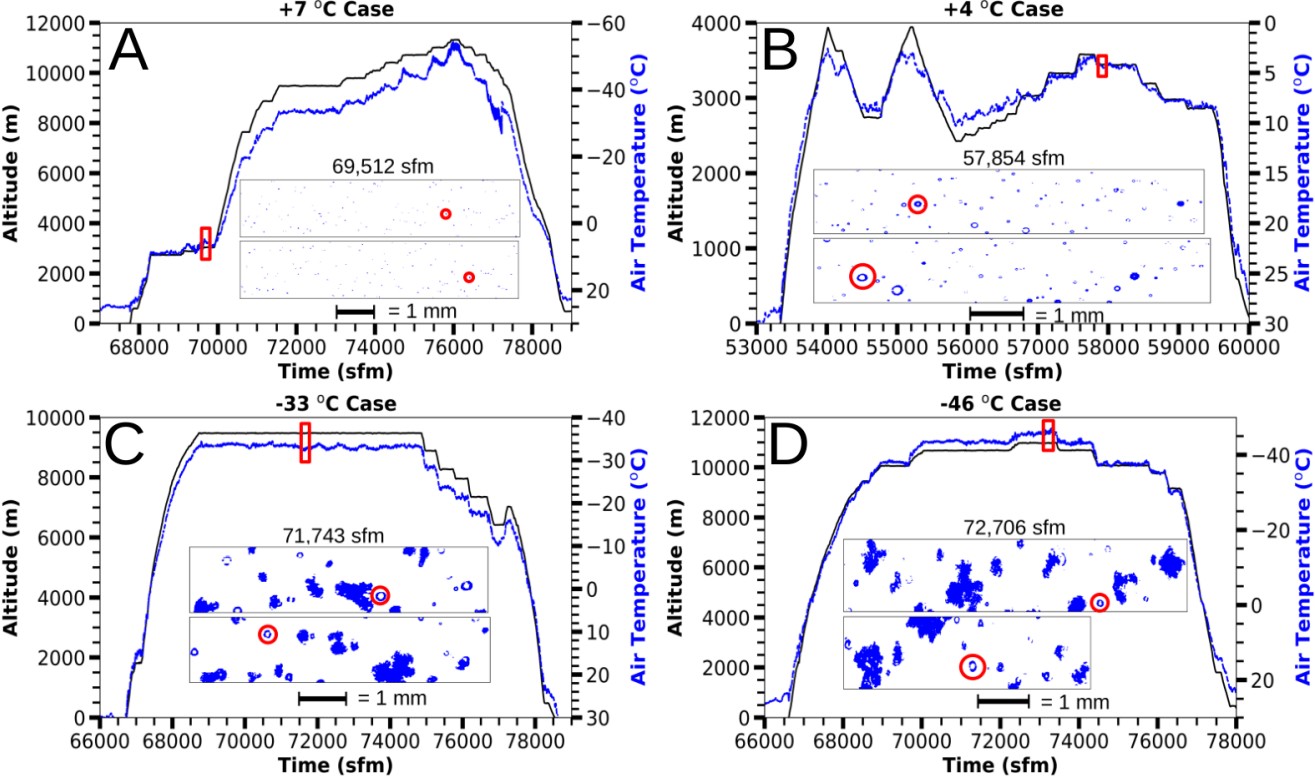

**Figure 4: Plots showing time series in seconds from midnight (sfm) Coordinated Universal Time (UTC) of altitude (solid, black) and air temperature (dashed, blue) for four analyzed flights. The red rectangles enclose the 60 s case study segments associated**
**with each flight. Details for each segment are given in Table 2. Center panels contain representative 2D-S images from within the analyzed cases. Red circles within the 2D-S images show examples of particles with Poisson spots that are discussed in the text. The upper, red circle marked particle for the -33 ºC case is approximately 180 µm in diameter.**

## 6 Results

Cloud probe images (2D-S and HVPS3) are processed and combined to create a 1 Hz ECP data set, which is used to derive
backscatter coefficients (Eq. (4)) and uncertainty (Eq. (5)). The absolute uncertainty is greatest for the +7 °C case (Fig. 5A) because the backscatter coefficient magnitude is 10 times the +4 °C case and 100 times the cold temperature cases. Figure 6A shows that OID and ECP backscatter coefficients agree (within 3 standard deviations) at all times for the +7 °C case. The greatest difference occurs at 10 s, where the ECP backscatter coefficient is over twice the OID's. Overall, the +7 °C case ECP backscatter coefficient is typically higher than the OID. The +7 °C water cloud contains high concentrations (Fig. 7A)
of small droplets (i.e. 20 µm mean diameter), with the majority of the calculated backscatter due to droplets between 5 µm and 30 µm (Fig. 7D).

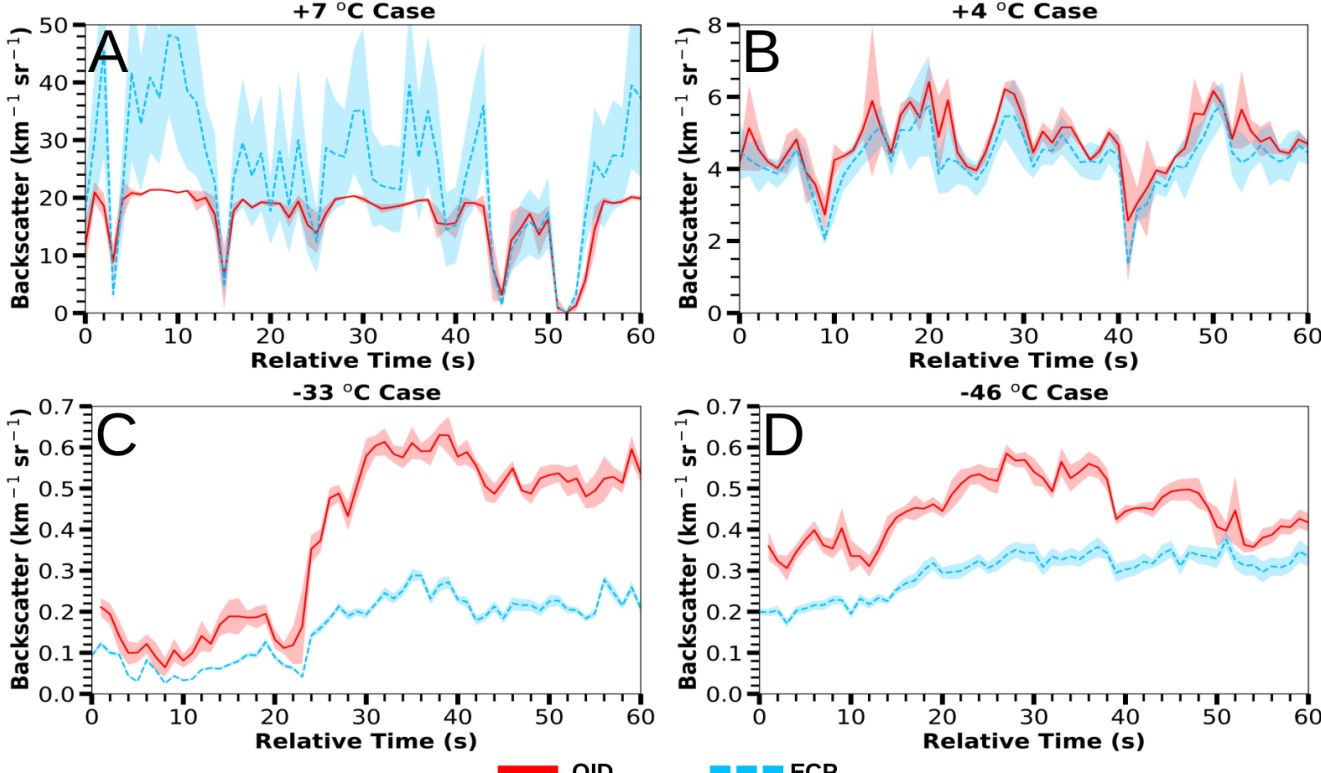

**Figure 5: Plots showing 1 Hz time series in seconds of backscatter coefficients with a shadowed range for the four case study segments (Table 2). Note the differing backscatter coefficient scales for the warm cloud cases. The external cloud probes (ECP)**
**shadowed range is the measurement uncertainty determined from Eq. (5) and represents 1 standard deviation. The Optical Ice Detector (OID) shadowed range is 1 Hz data representing 1 standard deviation computed of the 5 Hz measurements. The ECP backscatter coefficient is obtained from particle data processed with the area method.**



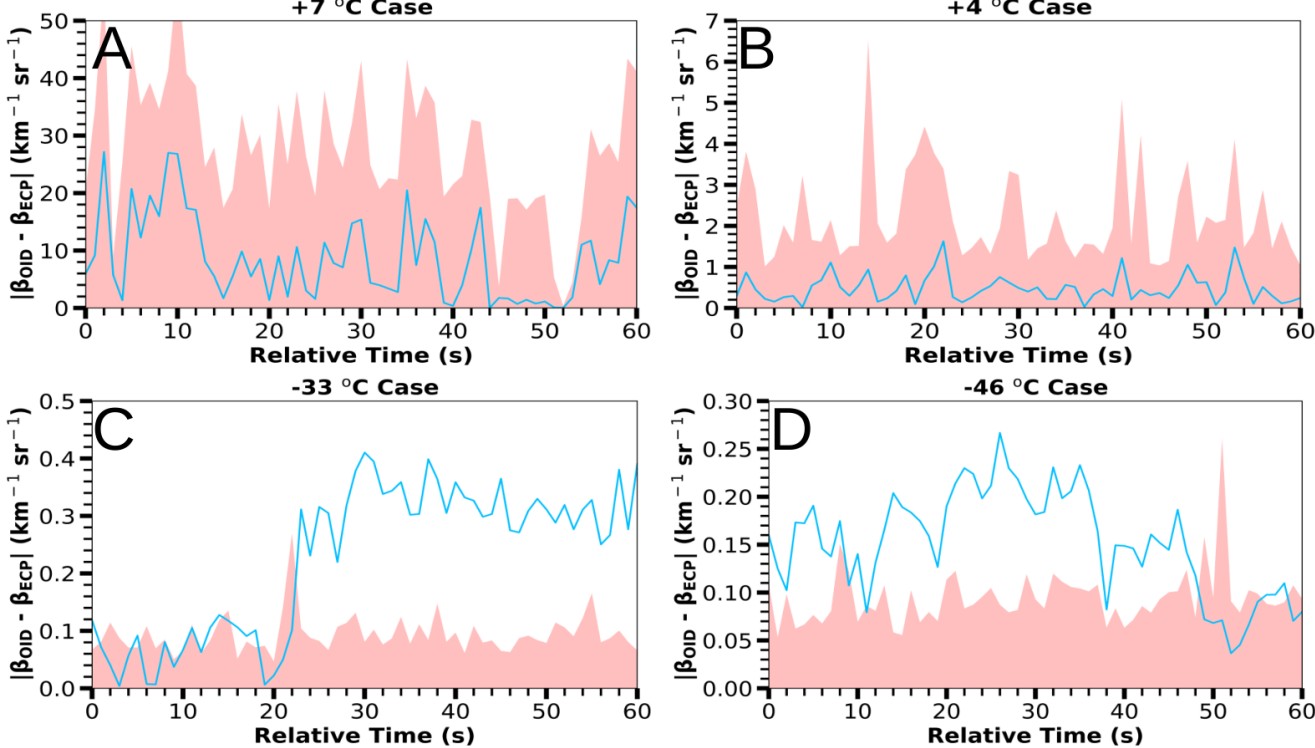

**Figure 6: Plots showing 1 Hz time series of the absolute difference (blue line) between the Optical Ice Detector (OID) and external cloud probes (ECP) backscatter coefficients for the four case study segments (Table 2). The shadowed region top represents 3 standard deviations of OID and ECP backscatter coefficients, determined using Eq. (7).**

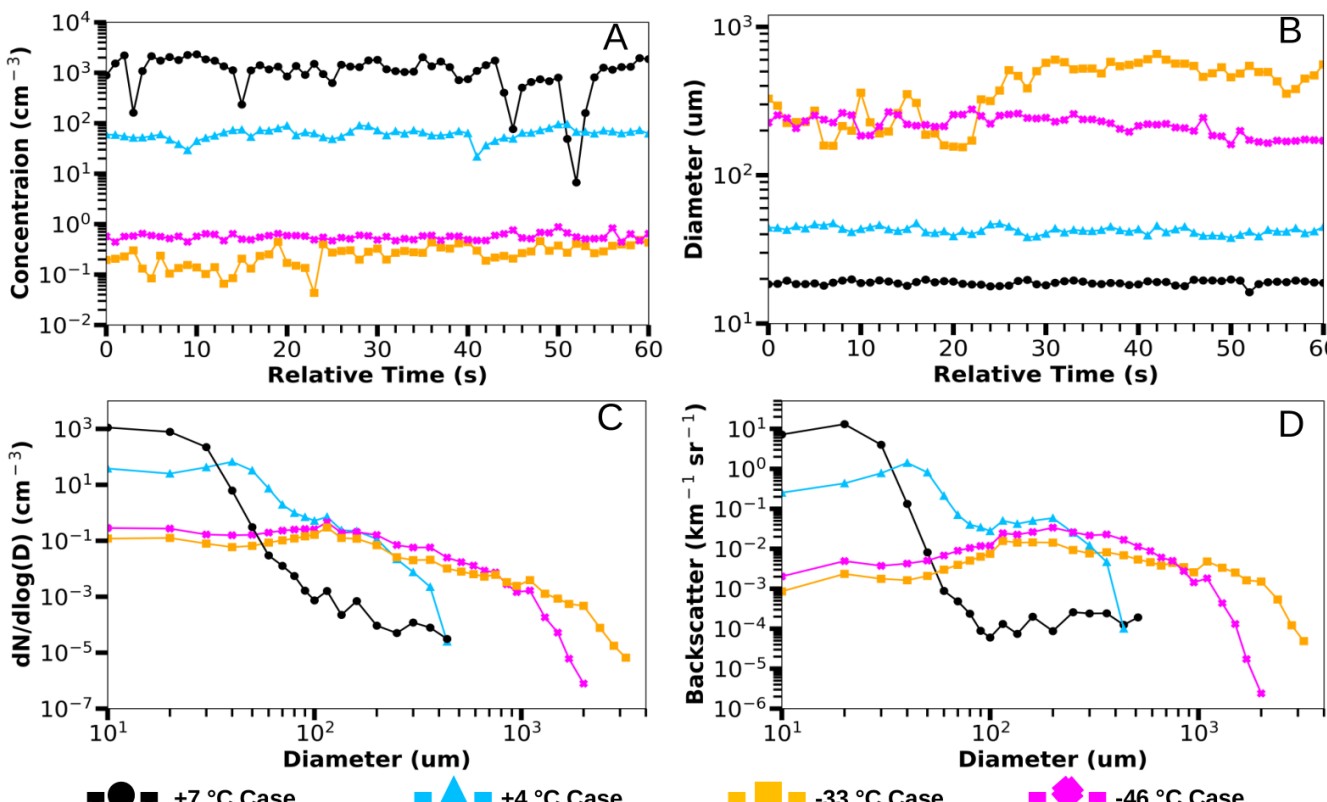

**Figure 7: Plots showing measurements within the 60 s segments of four analyzed cases (Table 2). The upper left plot (A) is total number concentration measured by the external cloud probes (ECP), specifically the Two-Dimensional Stereo (2D-S) and High-Volume Precipitation Spectrometer Version Three (HVPS3) probes (Table 1). The upper right plot (B) is mean particle diameter measured by the ECP. The lower left plot (C) is a log-log plot of number density function vs. diameter measured by the ECP. The lower right plot (D) shows ECP backscatter coefficient versus the particle diameter. The y-axis of plot D is not normalized by size channel width; however, neighboring channels typically have simple widths (see Table 1 for details). The ECP backscatter coefficient is obtained from particle data processed with the area method.**

Figure 5B shows that the +4 °C case has ECP backscatter coefficients that vary with the OID backscatter coefficient with the smallest difference between the two of any of the four cases; however, the ECP backscatter coefficient is consistently lower. The OID and ECP backscatter coefficients agree within uncertainties (Fig. 6B) throughout the entire period. The difference changes little because the particle concentration (Fig. 7A) and mean diameter (Fig. 7B) are approximately constant. Compared to the +7 °C case, the concentration is an order of magnitude less, and the mean diameter is greater, being approximately 50 µm instead of 20 µm. Many of the observed drops are from 30 µm to 70 µm (Fig. 7C), with the particle size distribution peaking at approximately 50 µm. Particles from 100 to 300 µm diameter are 100 times more plentiful than in the +7 °C case.

For the -33 °C case (Fig. 5C and Fig. 6C), the OID and ECP backscatter coefficients agree for the first 20 s. Similarly to the +4 °C case, the OID backscatter coefficient is consistently higher than that ECP backscatter coefficient. However, around 24


s there is a sudden change in backscatter coefficient agreement that corresponds to an increase in mean particle size and concentration. Figure 7A shows particle concentration increasing from approximately 0.1 cm⁻³ to 0.5 cm⁻³, and Fig. 7B shows the mean particle diameter increasing from 200 µm to slightly over 600 µm around the 24 s change. While both the OID and ECP backscatter coefficients increase at this time, the magnitude of the OID coefficient is significantly larger. For the last 30 s, the OID backscatter coefficient is 2 to 3 times the ECP backscatter coefficient, with disagreement exceeding 3

standard deviations. The particle size spectrum (Fig. 7C) and backscatter coefficient distribution (Fig. 7D) are much broader and flatter than in the warm cloud cases. Figure 7D indicates the majority of the backscatter coefficient is due to particles from 100 to 2000 µm in diameter.

The -46 °C case (Fig. 5D) also shows significant differences between the backscatter coefficients. For the first 48 s, the backscatter coefficient is in disagreement (Fig. 6D), with the ECP backscatter coefficient being consistently less than the

OID's. However, both backscatter coefficients exhibit the same temporal trends. For the last 12 s of the -46 °C case, the backscatter coefficients are mostly in agreement. As Fig. 7B shows, the mean particle diameter is nearly constant at approximately 200 µm, with a decrease at the end of the period. Both the -46 °C case and -33 °C case show 2D-S images with large, irregularly shaped particles (Fig. 4D and 4C, respectively). The -46 °C case has small particles that are irregular in shape, while the -33 °C case contains small particles that appear round.

There is a strong correlation (0.91) between $\beta_{ECP}$ versus $\beta_{OID}$ with the linear fit line being less than the one-to-one line over most of the data range (Fig. 8). The -33 °C case has higher OID backscatter coefficients compared to similar ECP backscatter coefficients from the -46 °C case. This difference at an ECP backscatter of 0.2 km⁻¹ sr⁻¹ (Fig. 8) is far greater than the calculated uncertainty, which indicates an unaccounted source of systematic error. Figure 9 shows the Nevzorov probe TWC and backscatter coefficient have a power-law relationship. The Backscatter-TWC relationship indicates the OID

instrument is sensitive to TWC over a wide range of cloud conditions, including clouds with TWC as low as 0.02 g m⁻³. The OID's sensitivity to small values of TWC (plot insert in Fig. 9) in optically thin ice clouds is an important characteristic for deployment on high flying aircraft. The OID has less scatter than the ECP for the Backscatter-TWC plots, likely due to the larger sample volume of the OID. Figure 10 includes all observations greater than 0 ℃ and less than -20 ℃ for the four case study flights. Data between 0 ℃ and 20 ℃ are omitted to exclude strongly mixed-phase cloud conditions as the ECP data

processing, and thus calculated backscatter coefficients, do not account for ice and liquid water backscatter efficiencies simultaneously. Including the whole flight, there is much more scatter between the ECP and OID backscatter coefficients (Fig. 10) than for the four 60 s flight segments (Fig. 8). Similar to the flight segment results in (Fig. 8), the trend over the whole flights (Fig. 10) indicates the ECP backscatter coefficients are lower than the OID coefficients. Also, there is a similar amount of scatter (Fig. 10) (0.74) between the OID backscatter coefficient and the Nevzorov probe TWC as seen in the flight

segments (Fig. 9).





**Figure 8:** The Optical Ice Detector (OID) versus external cloud probes (ECP) backscatter coefficients for four analyzed cases (Table 2). Each point is colored by case (see legend) and represents 1 s of data (Fig. 5) with bars indicating uncertainty. The least square fit is given by the black line (see equation in legend). The dashed line is a one-to-one correspondence for the ECP and OID data. The ECP backscatter coefficient is obtained using image data processed with the area-equivalent diameter method for all cases.

**Figure 9: Plots showing Nevzorov probe total water content (TWC) versus Optical Ice Detector (OID) backscatter coefficients (top) and external cloud probes (ECP) backscatter coefficients (bottom) with a logarithmic x-axis. Backscatter coefficients are** 400 **separated by warm (+7 °C and +4 °C) cases (square, black) and cold (-33 °C and -46 °C) cases (round, blue) (Table 2). Insets in the top-left corners show further detail of Nevzorov probe TWC versus respective backscatter coefficients for the -33 °C and -46 °C cases. Each point represents 1 s of Nevzorov, OID, and ECP data. The ECP backscatter coefficient is obtained using image data processed with the area-equivalent diameter method for all cases.**

**Figure 10:** The external cloud probes (ECP) backscatter coefficients versus Optical Ice Detector (OID) backscatter coefficients (top) and Nevzorov probe total water contents (TWC) versus OID backscatter coefficients (bottom) for all times within the four flights (Table 2) that contain the four temperature cases. The 1 Hz data is grouped by cold (round, blue) and warm (square, black) environments. Data are excluded when temperatures are between 0 °C and –20 °C. Liquid scattering efficiencies are used for temperatures greater than 0 °C and ice scattering efficiencies used for temperatures below –20 °C. OID data has a 20 km⁻¹ sr⁻¹ limit to avoid times where observations are above the OID's detection limit. The lower plot's red dashed line indicates the fit equation in Fig. 9. The ECP backscatter coefficient is obtained using image data processed with the area-equivalent diameter method for all cases.

## 7 Discussion

The +7 °C case has the largest backscatter coefficient and the only case where ECP derived backscatter coefficient is larger than the OID measurement. This is due to high droplet concentrations saturating the OID above its upper response limit of



22 km$^{-1}$ sr$^{-1}$. The OID's limited dynamic range in a dense cloud of small water droplets is not a serious liability since sensing ice particle concentration is the main measurement objective and warm water clouds do not produce ice particle icing. Furthermore, environments containing severe conditions would not require such large concentrations to be flagged by the OID. Note that the cumulus cloud sampled in the +7 °C case has a large water content (Table 2), and the backscatter

coefficients to water content comparisons indicates the OID is able to measure over 1.0 g m$^{-3}$ before saturation (Fig. 9).

Excluding the +7 °C case, ECP backscatter coefficients are consistently lower than those measured by the OID (Fig. 5B, 5C, 5D, and summarized in Fig. 8). The consistent difference between the OID and ECP suggests a possible systematic error. Two-dimensional cloud imaging probes (Fig. 1) have several factors that can bias sizing and counting of particles. Particle sizing uses the area-equivalent diameter method for all cases, which is appropriate for liquid droplets; however, this may not

be the best sizing method to obtain the equivalent diameter for ice particles using Mie backscattering. The fast-circle diameter method results in larger diameter ice particles and larger calculated backscatter coefficients from the ECP (see supplemental material). Particle diameters less than 100 µm are difficult for the 2D-S to measure due to the small depth of field, which could result in lower backscatter coefficients than those measured by the OID. However, the 2D-S LWC is larger than the Nevzorov probe LWC measurements for the CAPE2015 field project. Since the OID backscatter coefficients

are more reflective of the 2D-S LWC, the Nevzorov LWC may be low due to an inability to fully capture and/or evaporate the largest water droplets. OID measurements may have a bias despite no indication of malfunctions. Another possibility is the OID receiving backscatter from particles too small to be measured by the two-dimensional cloud imaging probes; however, Fig. 7D indicates low backscatter coefficient contribution from the smallest sized channels. Additional measurements with carefully calibrated instruments are necessary to resolve the discrepancy between the OID and ECP for

ice containing clouds; however, the OID and ECP measurements do agree within the calculated uncertainties for liquid clouds (Fig. 6).

There are other possible errors which affect both liquid water and ice water measurements. A bias in concentration could be due to an error in air speed used for determining the sampling volume (Eq. (2)). However, the aircraft's air speed error is estimated to be between 1 % and 3 % based on measurement comparisons, which is too small to account for the bias.

Another error source may be coupling probes to the ambient airflow environment. Air flow deviations around instrument pylons can influence particle concentrations even when probes are placed well in front of the wing's leading edge (Baumgardner, 1984). Pressure changes and streamlines alter particle flow around both wings and instruments, with small particles being most heavily affected (Spanu et al., 2020). This phenomenon results in lower measured concentrations of small particles compared to actual values in undisturbed air. Another possible concentration bias is in processing

asynchronous 2D probe images to calculate the sample area. Errors in determining sample area are size dependent and larger for small particles (Korolev et al., 2013a); therefore, such errors are more important for the +7 °C and +4 °C cases.



A sample area error affects particle concentration, which has the larger uncertainty contribution. The +4 °C case (Fig. 9) has an average contribution to uncertainty (Eq.(6)) of 0.4 km$^{-1}$ sr$^{-1}$ from particle concentration, while only 0.045 km$^{-1}$ sr$^{-1}$ from particle size. Similarly, the -46 °C case (Fig. 5D) has uncertainty of 0.24 km$^{-1}$ sr$^{-1}$ from concentration and only 0.03 km$^{-1}$ sr$^{-1}$

from particle size. Hence, the primary contributor to total uncertainty is fluctuation in concentration. Concentration uncertainty can be reduced by averaging over longer time periods; however, systematic differences between the OID and ECP would not change.

Bias in calculated sample volume cannot explain the change in agreement between OID and ECP backscatter coefficients that occurred between 20 s and 30 s for the -33 °C case (Fig. 4C, 5C, and 6C), or the overall disagreement in the -46 °C case.

Onset of the -33 °C case discrepancy could be due to a change in the particle size spectrum (Fig. 11). Manually reviewing 2D-S and HVPS3 images indicates an increased number of larger particles between the first and last 20 s of the -33 °C case. This change in particle size distribution is believable since there is a smooth decrease in concentration with increasing size. Furthermore, the 2D-S/HVPS3 particle size distribution is similar to the Two Dimensional Cloud (2D-C) probe (Fig. 2) distribution (Fig. 11), and performance checks conducted by the manufacturers after CAPE2015 found no measurement

issues. Additionally, a manufacturer's review of the OID indicated no measurement issues. Therefore, both the OID and ECP measurements seem to be valid for the -33 °C case.



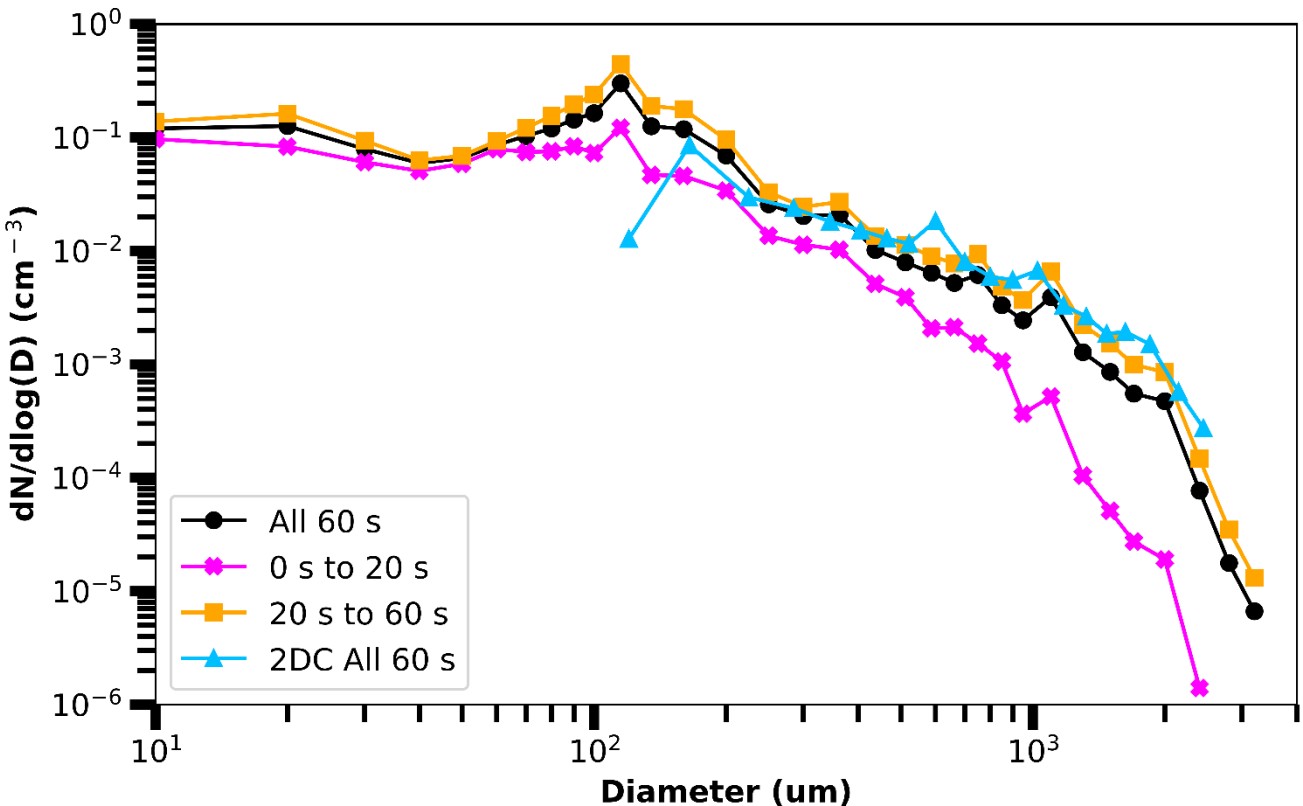

**Figure 11: The size spectrum normalized by bin width measured by the external cloud probes (ECP) for the -33 °C case (Table 2). The spectrum of the entire 60 s case (round, black) has been separated according to times of more (cross, magenta) and less (square, orange) agreement between ECP and Optical Ice Detector (OID) backscatter coefficients. Also given is the 2D-C spectrum (triangle, blue) obtained using the fast circle and full particle reconstruction processing methods. The 2D-S and HVPS3 diameters for the ECP combine spectrum are obtained using image data processed with the area-equivalent diameter method.**

A notable difference between the -33 °C and -46 °C cases is LWC measured by the Nevzorov probe (Fig. 12). The Nevzorov probe measurement for the -46 °C case has near zero LWC, as expected for a temperature below the threshold temperature for homogeneous freezing. However, the Nevzorov probe measurement indicates a mixed phase cloud for the -33 °C case. While there is LWC data shown for the entire -33 °C case, LWC increases from an average of 0.022 g m⁻³ to 0.027 g m⁻³ (~30 % of the TWC) just after disagreement between backscatter coefficients begins. An abrupt increase in TWC also occurs at this time, which should result in both the OID and ECP backscatter coefficients increasing; however, while the increase does occur simultaneously (Fig. 5C), the discrepancy between backscatter coefficients after the increase is of interest. While the increase of LWC is small, the additional LWC would increase the difference between backscatter coefficients since ECP processing assumes an ice backscatter efficiency, which is lower than water backscatter efficiency (Fig. 4). Furthermore, the increase in LWC likely corresponds to the particle size distribution increase in the 100 to 200 μm range (Fig. 11), which has a high contribution to the overall backscatter coefficient (Fig. 7D). It should be noted that while the Nevzorov probe has a





collection efficiency of nearly 1 for droplets with a diameter less than 100 µm (Korolev et al., 1998), collection efficiency is

approximately 0.5 for clouds with a mean particle volume diameter of 150 to 200 µm (Biter et al., 1987). During the

disagreement period, there are significantly more particles with diameters from 150 to 200 µm (Fig. 11), which would

require a greater, and not readily available, correction to Nevzorov probe LWC measurements.

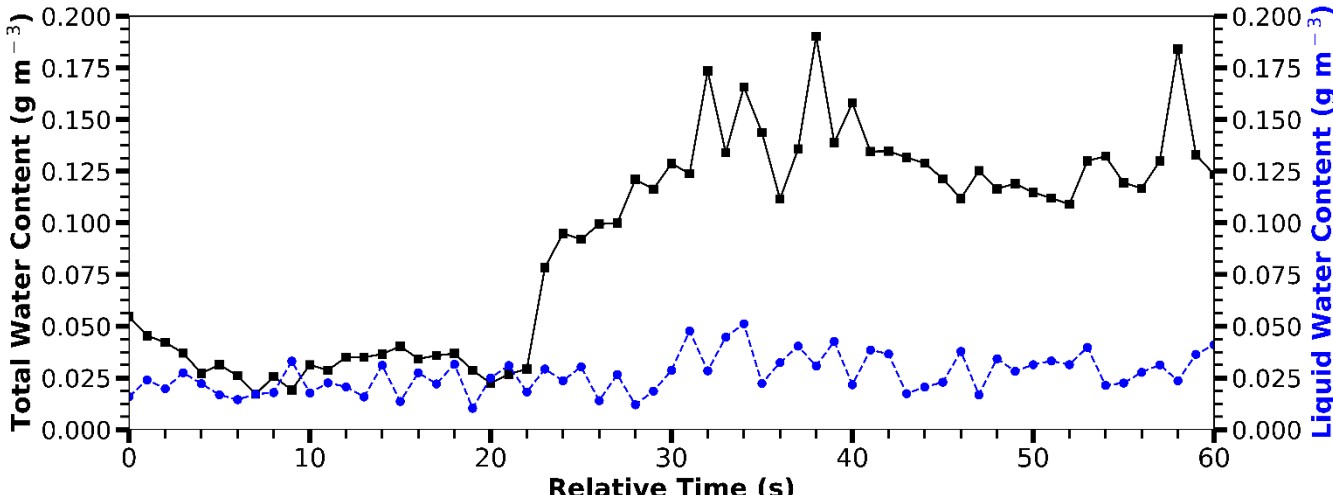

**Figure 12: Nevzorov probe measured total water content (TWC) (solid, square black) and liquid water content (LWC) (dashed, round blue) for the -33 °C case (Table 2). Each point represents 1 s of Nevzorov data. The average LWC is 0.022 g m⁻³ ± 0.009 g m⁻³ and the average TWC is 0.032 g m⁻³ ± 0.008 g m⁻³ during times of Optical Ice Detector (OID) and external cloud probe (ECP) backscatter coefficient agreement (0 s to 20 s). The average LWC is 0.027 g m⁻³ ± 0.009 g m⁻³ and the average TWC is 0.092 g m⁻³ ± 0.05 g m⁻³ during times of OID and ECP backscatter coefficient disagreement (30 s to 60 s).**

Even at temperatures as low as -33 °C, supercooled liquid water droplets can still exist (Rosenfeld and Woodley, 2000).

Supercooled liquid water droplets have even been observed by the same OID at temperatures as low as -30 °C in Anderson

and Ray (2019). Similarly to Anderson and Ray (2019), such supercooled liquid water droplets are evident, but not proven,

by the round particles seen in 2D-S images taken at the LWC measurement peak (Fig. 4C). Additionally, the Rosemount

Icing Detector (Fig. 2) rod frequency (not shown) indicates the presence of LWC. The authors speculate that the presence of

supercooled liquid water droplets in the -33 °C case results in larger OID backscatter coefficients due to higher (see Q Ratios

in Table 1) liquid water backscatter efficiency (Fig. 3). Calculating ECP backscatter coefficients using liquid water scattering

efficiencies for particles less than 500 µm and ice scattering efficiencies for particles larger than 500 µm, results in ECP/OID

agreement for the -33 °C case, and would eliminate the systematic difference at ECP backscatter coefficient of 0.2 km⁻¹ sr⁻¹

in Fig. 8. Accounting for mixed phase conditions in ECP backscatter coefficients would require a methodology able to

simultaneously utilize a particle size spectrum for both liquid water and ice, which is beyond the scope of this study. The

authors would like to note that mixed phase clouds are only one possibility, and further analysis is required.



While each of the previously mentioned sources of error increase uncertainty within the backscatter coefficient comparisons and should be acknowledged, their contributions are small relative to the assumption made of particles either being water or ice. Incorrect phase assumptions when determining the backscatter efficiencies used in calculating backscatter coefficients can have significant impact on the results due to large differences in efficiencies between ice and water, particularly for

particles over 140 μm in diameter (see Table 1). At a diameter of 1000 μm, incorrectly assuming water over ice (or vice versa) would impact the backscatter efficiency (and thus the backscatter coefficient) by 1500 %. This could be solved via testing in a closed system where particle phases can be known with certainty.

For ice particles specifically, the most significant uncertainty is likely the processing method used to obtain the size of particles; i.e. area (figures presented herein) versus fast circle processing (figures presented in supplemental material). The

area processing method is best for spherical droplets since the imaging probes directly measure area; however, the best processing method for ice particles is not clear as a sphere is required that equivalently backscatters light, which likely is not an area equivalent sphere obtained from one viewing angle.

## 8 Conclusions

A backscatter coefficient comparison is conducted for both warm and cold cloud conditions using an in-situ Lidar (OID) and

observations from externally mounted cloud particle probes (ECP). ECP backscatter coefficients are calculated using combined 2D-S and HVPS3 data over the 5 to 30 000 μm size range. ECP derived backscatter coefficients are consistently lower than OID measurements for the +4 °C, -33 °C, and -46 °C cases, suggesting a systematic bias in either or both data sets (Fig. 5). However, the ECP derived backscatter coefficient is higher than OID measurement for the +7 °C case due to OID saturation. In the +7 °C and +4 °C cases, the particles that primarily contribute to the backscatter coefficient are in the 5

to 50 μm diameter range, while in the -33 °C, and -46 °C cases, the particles that contribute the most are in the 200 to 3000 μm diameter range (Fig. 7D). ECP backscatter coefficients are less than 3 standard deviations from OID measurements, indicating agreement for the +7 °C case and +4 °C cases (Fig. 6). However, there is disagreement in the majority of the -46 °C case. The -33 °C case has disagreement for 36 s out of 60 s, which is possibly due to using ice scattering efficiencies for all particles when small particles are potentially liquid. All case comparisons show qualitatively that backscatter coefficients

are a suitable proxy for LWC and IWC as low as 0.02 g m⁻³ (Fig. 9). A comparison between OID and ECP backscatter coefficients to Nevzorov probe TWC for all times during four CAPE2015 flights (Fig. 10) has a correlation above 0.7. Considering the measurement uncertainties, the case study analysis, and overall flight comparison, the OID is a useful tool for detecting dangerous ice particle icing conditions, even at low concentrations.

The CAPE2015 field project analysis could be expanded to include additional field projects. Additionally, future work could

include both the 905 nm and 1550 nm OID wavelengths. Using both wavelengths would allow two parameters of cloud

particle size distributions to be determined, helping to obtain a TWC. Additionally, OID polarization measurements could distinguish the liquid/ice fraction of clouds (Yang et al., 2003). New instrumentation, such as the Particle Habit Imaging and Polar Scattering (PHIPS) probe (Schön et al., 2011) deployed during the CapeEx19 Florida field project, would be very useful in determining the fraction of liquid water in mixed phase clouds. Quantifying the liquid water fraction of mixed

phase clouds would allow for creation of both liquid and ice 1 Hz particle size distributions, so liquid and ice scattering efficiencies could simultaneously be used to obtain the backscattering coefficient. PHIPS data would also aid in the identification of ice particle habits. Knowing the ice particle habits would allow for more accurate scattering calculations to be utilized (Yang et al., 2005).

**Code Availability**

ADPAA software used to process the raw data is freely available from a software repository (Delene et al., 2020). The SODA2 software used for image probes processing is from an open software repository (Bansemer, 2013). Specialized software used in the paper's analysis is available in a publicly accessible repository (Wagner and Delene, 2020b). Software processing configuration details are documented in a work-flow script entitled "oid_analysis_workflow" (Wagner, 2020). The work-flow script contains execution calls to ADPAA modules used to process and analyze the paper's data set. Whereas

ADPAA processes data from many field projects, the work-flow script applies only to this paper's analysis. The MiePlot software package (Laven, 2018) used to calculate backscatter efficiencies is readily available online.

**Data Availability**

The paper's data set is freely accessible online through the Chester Fritz Library's data collection (Wagner and Delene, 2020a). Archived data consists of 1 Hz data from four Florida flights in 2015. Data set measurements include the Two-

Dimensional Stereo probe (2D-S), High-Volume Precipitation Spectrometer Version Three (HVPS3) probe, Nevzorov probe and Optical Ice Detector (OID). 2D-S and HVPS3 spectrum are combined to create a single particle size distribution spanning both probe size ranges. Backscatter coefficients are computed from this composite size spectrum. Atmospheric variables such as pressure, temperature, dew point temperature, and wind velocity are also included in the data archive.

**Author Contribution**

First author, Shawn Wagner, conducted data analysis and led manuscript writing. Co-author David Delene developed the project's design and scope. David Delene headed external cloud probe analysis. Both authors reviewed and edited the manuscript.

**Competing Interests**

The authors declare that they have no conflict of interest.





## Disclaimer

Opinions expressed in this paper are those of the authors and do not reflect opinions of others who aided in this research.

## Acknowledgements

We are grateful to the National Center for Atmospheric Research (NCAR), and the lead developer Aaron Bansemer, for making available the SODA2 software used in this research. Thank you to NASA for use of 2D-S, HVPS3, and Nevzorov probe instruments during the CAPE2015 field project. Kaare Anderson provided processed OID data for CAPE2015. The CAPE2015 aircraft flight crew members included flight engineers Jamie Ekness (Graduate Student), Kryzsztof Markowicz (Graduate Student) and Nicholas Gapp (Undergraduate Student), flight scientist David Delene, and research pilots Wayne Schindler and Jonathan Sepulveda. Thanks to Andrew Detwiler for providing comments on draft versions of the paper. Thank you to Neil Nowatzki of University of North Dakota's Academic Support Services for help with creation of Fig. 1. Additionally, we acknowledge anonymous reviewers and thank them for the time they devoted to reviewing the manuscript. Funding to conduct data analysis was provided by the North Dakota Department of Commerce (University fund number UND0021191).

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
