# Peer review of "Technique for comparison of backscatter coefficients derived from insitu cloud probe measurements with concurrent airborne Lidar"

_Atmospheric Measurement Techniques, 2022_

## Author Comment (AC3)

**Amt-2022-87 Authors' Initial Response to Anonymous Referee #3 14 July 2022**

The authors would like to thank referee 3 for taking the time to write an extensive and well composed review of the manuscript which provides thought provoking suggestions and questions. Replies to the line specific comments are given below in italic after repeating the reviewer's comment.

**Major Comments:**

**Backscatter coefficient derivation from ECP data**

First of all, the assumption of ice sphere for the derivation of the backscatter coefficients from ECP data would cause a significant systematic bias. As confirmed by many aircraft observations of ice crystals, a majority of ice crystals have nonspherical shapes. The scattering properties of nonspherical ice crystals differ significantly from those of spherical ice. Although the authors claim that the uncertainty of the backscattering coefficients associated with the spherical ice assumption is much less than the uncertainty from particle size distribution (PSD) measurements (*Line 234–236*), these uncertainties would involve systematic biases, which will be carried over computing the backscattering coefficients from the PSD measurements.

To improve the validity of the present analysis for ice clouds, I suggest the authors add the following analysis to the ice cloud cases. To convert the extinction coefficients from backscattering coefficients through the inversion process, we use the lidar ratio (S; extinction-to backscatter ratio) of ice clouds that is empirically determined for each hydrometeour, and the latest value of the lidar ratio for ice clouds is, for example S = 32 sr at 532 nm (Holz et al., 2016). The lidar ratio at 905 nm could differ slightly from the one at 532 nm due to a slight difference of the real part of the ice refractive index between these wavelengths, but it should be quite consistent. The authors are strongly encouraged to perform the additional ECP data analysis with a lidar ratio of 32 sr for ice cloud cases.

Also, I noticed that the backscattering efficiency is introduced in Line 238 without a definition. Please clearly define the backscatter efficiency in the corresponding sentence. I believe that the authors defined the backscattering efficiency of a single particle as

$$Q_{back} = \frac{Q_{ext}\omega P_{11}(\pi)}{4\pi},\tag{R1}$$

where  $Q_{ext}$  is the extinction efficiency;  $\omega$  is the single-scattering albedo; and  $P_{11}(\pi)$  is the scattering phase function at 180° degree. I would like to clarify if the authors include a denominator of  $4\pi$ . If the above is correct, the lidar ratio can be described as

$$S = \frac{Q_{ext}}{Q_{back}},\tag{R2}$$

Otherwise, I am wondering if the above definition differs from what is actually defined because the paper states that the fourth Stokes component V is the focus of the study (Line 97).

It is true irregular shaped ice crystals scatter differently than spherical ice crystals, and clouds have irregular shaped ice crystal as shown in the inserts of Fig. 4. There are two size parameters (radius/diameter) that are important: the radius in Eq. 4 of the paper for deriving backscatter coefficients and the diameter of the scattering efficiency at 180 degrees (Fig. 3).

Typically these are the same; however, it is not apparent that they would be for non-spherical particles.

First, note that the uncertainties for determining particle diameter from 2-dimensional images are large (see the uncertainty analysis within the paper). The main manuscript uses an area-equivalent diameter, and the supplemental material uses a fast-circle diameter. The area-equivalent diameter is typically more acceptable since it is the equivalent diameter of a sphere which has the same area as that observed within the 2-dimentational probe images. Hence, when converted to a radius, squared, and multiplied by  $\pi$  in Eq. 4, it provides the area of the imaged particle. There could be biases that are important to acknowledge in obtaining the area of a particle from a 2-dimentional image if the particles are not randomly oriented either in the probe images or in the Lidar backscatter volume. We will add an additional note on this potential bias in the manuscript around line 295.

There is also the diameter in the backscatter efficiency equation (Figure 3), which is currently the same as the diameter used in Eq. 4; however, since non-spherical and spherical particles do not scatter the same, a different diameter may be better to use as the equivalent backscatter diameter than the area equivalent diameter for obtaining the backscatter efficiency. Additionally, it is not clear that the backscatter coefficient in Eq. 4 should be related to the area equivalent diameter since the backscatter is dependent on surface waves, and the interference of the surface waves (see comment/discussion of Referee 1).

In conclusion, this manuscript uses the most accept diameter for ice particles, and the supplement presents the fast-circle diameter for comparison, which we believe is the best that can be done. We will add a sentence or two related to this topic in the discussion section of the paper. While it is beyond the scope of this article, future work could use the ECP/OID data set to determine the diameter that provides the best agreement with the Lidar measurement for ice clouds.

The authors apologize as it was believed to be clear in the manuscript that a lidar ratio is not assumed for the calculations. The OID measures attenuated backscatter at different ranges and then fits a curve to the exponential function in Eq. 1. Since the lidar only samples along the wing of the aircraft (see Fig. 2), the cloud is homogenous enabling the curve fit. The attenuation is alpha, and the true, unattenuated backscatter is the y-intercept of the curve from the data fit. The lidar ratio is then derived from these two measured values. Therefore, the lidar ratio changes depending on the cloud conditions, and no single value is applied specifically for ice cases. Hence, the aircraft lidar processing is different than satellite based lidar processing where clouds sampled are not homogenous along the lidar beam at different range gates. We will include this information in the manuscript beginning at line 136.

This definition of the backscattering efficiency is correct, and a denominator of  $4\pi$  is included. This equation will be added to the updated manuscript at line 245 as Eq. 5.

**Backscatter coefficient derivation from OID data**

As seen in Eq. (1) in the manuscript, the lidar signals from a certain location of ice clouds relative to the location of the aircraft can be attenuated by ice crystals in between the two locations. Therefore, the lidar signals need a correction with the two-way transmissivity to obtain the backscatter coefficient. The authors cite Lolli et al. (2013) for the extinction coefficient inversion for the present analysis, this paper is for rain droplets and the predefined lidar ratio for rain droplet (i.e., 50 sr in Lolli et al., 2013) may be inaccurate for small liquid droplets and ice clouds. Please add a few sentences describing how the extinction efficiency is derived for both liquid and ice cloud cases. In particular, what lidar ratios are used to estimate the extinction cross-section through the inversion of lidar measurements for each ice and liquid cloud case?

The lidar ratio is derived by curve fitting the OID measured attenuation and the unattenuated backscatter, allowing for the lidar ratio to vary with the changing cloud conditions. As the sampling volume is relatively small, limited to only 10 m from the aircraft, attenuation by ice crystals is very limited, so a correction with the two-way transmissivity is not necessary. The authors apologize that this is not clear in the text, and this information will be added in the following manuscript beginning at line 136.

**Minor comments:**

1. "Hulst (1981)" should be "van de Hulst (1981)" throughout the manuscript.

This is an error by the authors and will be corrected in the following manuscript version.

2. Lines 96–97 "*The 905 nm bean enables measurement of the fourth Stokes parameter (V) (Liou and Yang, 2016; Hulst, 1981) and is the focus of the study, …*" I got an impression that the lidar instrument measures only the fourth component of the Stokes vector (*V*) from the manuscript. However, it actually measures the first component of the Stokes vector (*I*) in addition to the fourth component according to Ray and Anderson (2015), doesn't it? Please clarify it.

It is correct that the OID measures both Stokes vector (I) and (V), however (I) was neglected to be mentioned as the results of (V) are the focus of this study. For clarity line 96 will be updated using "With the 905 nm the OID is able to measure both the first Stokes parameter (I) as well as the fourth Stokes parameter (V) (Liou and Yang, 2016; van de Hulst, 1981). The fourth Stokes parameter (V) is the focus of this study, and the 1550 nm wavelength channel is not used.

3. Lines 155-156 "*Images are produced when at least one array element is "shadowed" (i.e., reduced in intensity by 50% or more).*": Is there any reference that discusses the accuracy of estimated particle area from 2D-S with this approach?

McFarquhar Et al. (2017) discusses this point on page 15, and Figure 11-7 using a CIP probe. Their results indicate that setting a threshold of 50 % results in derived particle diameters 100  $\mu$ m less than those derived using a 70 % threshold, with this difference increasing with smaller sized particles. This citation will be added at line 156.

4. Lines 187–190, Eq. (2): Use the italic font for scaler variables in the main text to be consistent with Eq. (2).

This is an error by the authors and will be corrected in the following manuscript version.

5. Line 226 "geometric" should be "geometric optics".

This is an error by the authors and will be corrected in the following manuscript version.

6. Line 234-236 "While Mie theory ... ": Cairo et al. (2011) states in Page 561 that "Generally speaking, aspherical scatterers depress the forward and back- ward scattering and enhance the side scattering with respect to surface equivalent spheres, so an overestimation of the backward scattering may be expected when using Mie codes. An educated guess of such overestimation can be provided by looking at studies comparing the phase function of aspherical vs spherical scatterers, which suggest an average overestimation of the Mie backscattering coefficient by a factor 2, which may possibly get as large as a factor 4 or more, depending on particle sizes and shapes (Mishchenko et al., 1996)." Please revise the corresponding sentence to be consistent with the statement by Cairo et al. (2011).

To be more consistent with Cairo et al. (2011), line 234 will be updated in the following manuscript to "As Mie theory strictly applies to spherical particles, previous work has found that aspherical particles tend to have enhanced side scattering compared to spherical scatterers, resulting in Mie codes producing larger backscatter coefficients by an average of 2 or more (Cairo et al., 2011). However, this uncertainty is less than those associated with measurements of cloud particle sizes.

7. Lines 246-247 "an equivalent sphere": This should be clearly stated as "a projected area equivalent sphere" in order to avoid confusion with a volume equivalent sphere. The backscattering coefficient is proportional to the cross-sectional area of a particle for large size parameters (i.e.,  $Qext. = \sim 2$ ), so that the use of projected area equivalent spherical radius is relevant for both liquid and ice particles in the present analysis.

The authors respectfully disagree that this should be considered as a projected area equivalent sphere, as the surface waves mentioned in line 265 prevent  $Q_{back}$  from reaching an asymptote and becoming proportional to the cross-sectional area of the particle. Instead, a more appropriate labeling in line 246 would be "a backscatter equivalent sphere".

8. Lines 267–268: This statement is inconsistent with Lolli et al. (2013) that use a predefined extinction-to-backscatter ratio (or the lidar ratio) to estimate the extinction coefficients from lidar signals. Thus, the backscattering efficiency is necessary for to interpret OID data (as clearly indicated in Eq. 1).

In Eq. 1, an assumed lidar ratio is not necessary. The OID measures attenuated backscatter at various ranges and then fits this curve to the exponential function in Eq 1. The

attenuation is alpha, and the true unattenuated backscatter is the y-intercept of the curve. The lidar ratio is then derived from these two measured values. Thus, while the backscatter efficiency is necessary to calculate the ECP backscatter coefficients, it is not necessary for understanding the OID data.

9. Figure 3 caption: 0.0001 µm should be 0.0001 mm. Also, 3 µm should be 3 mm.

The values 0.0001  $\mu$ m and 3  $\mu$ m are intentional as they are to indicate the intervals between diameters used for the individual backscatter efficiency calculations, not the intervals in diameters between the average backscatter efficiency calculations. This is not clear in the manuscript. For the following manuscript, line 272 could be updated to "Intervals between diameters used for the individual backscatter efficiency calculations range from 0.0001  $\mu$ m to 3  $\mu$ m."

10. Figure 5a: What lidar ratio is used to derive the extinction coefficient for the OID analysis for this case? As Lolli et al., (2013) use a lidar ratio of 50 sr for rain drop that is significantly larger than those of cloud liquid droplets (~20 sr), the two-way transmissivity could be overestimated, so that OID derived backscattering coefficient might be underestimated. The authors are encouraged to clarify this.

As mentioned previously in comment 8, a lidar ratio is not assumed but rather repeatedly derived by using the measured attenuation and the true unattenuated backscatter in a curve fit. Thus, the lidar ratio changes to match the environment which encounters little attenuation due to the relatively small sampling volume.

11. Line 378 "..., which indicates an unaccounted source of systematic error.": I think this may be due to the backscattering efficiency bias associated with an ice sphere assumption. In Lines 295-296 the manuscript says "Eq (7) does not include systematic errors (e.g., uncertainty in backscatter efficiency)". I suggest the authors to mention that one of unaccounted errors would be a systematic bias in backscatter efficiency.

While a backscattering efficiency bias likely has an effect to some degree, as mentioned in comment 6 and in Cairo et al. (2011) calculated backscatter coefficients using Mie theory code on aspherical particles should be a factor of at least 2 higher than those that are measured. Thus, the backscatter efficiency bias cannot fully explain why the OID backscatter coefficients are higher than those calculated from ECP data. However, this bias is still important to acknowledge and to emphasize this face line 295 can be updated in the following manuscript to "The third standard deviation is used as a threshold in accordance with the Three Sigma Rule (Pukelsheim, 1994) since the OID is in development and Eq. (7) does not include systematic errors (e.g. systematic bias in the backscatter efficiency due to spherical particle assumptions)."

**References**

- Holz, R. E., and Coauthors, 2016. Resolving ice cloud optical thickness biases between CALIOP and MODIS using infrared retrievals. Atmospheric Chemistry and Physics, 16(8), 5075-5090.
- McFarquhar, G. M., and Coauthors, 2017: Processing of Ice Cloud In Situ Data Collected by Bulk Water, Scattering, and Imaging Probes: Fundamentals, Uncertainties, and Efforts toward Consistency. *Meteorological Monographs*, **58**, 11.1-11.33, https://doi.org/10.1175/AMSMONOGRAPHS-D-16-0007.1.

---

## Author Response (AR1)

**Amt-2022-87 Authors' Responses to Anonymous Referees**

The authors would like to express their appreciation and gratitude for input and suggestions of the three anonymous referees. The comments were clearly genuinely and heavily contemplated to provide excellent insight into areas of the manuscript which needed improvement. The original referee comments are presented first, followed by the author response in italic font and then the respective changes to the manuscript.

**Responses to Anonymous Referee #1**

(132) "The backscatter coefficient is calculated...". This β has contribution both from molecules and particles. Given the very high particle β measured, the molecular contribution could be neglected but have to be mentioned.

*It is true that the molecular backscatter contribution to β is negligible in comparison to the particles. From Anderson et al., 2015: "The OID is not sensitive enough to measure molecular scattering that is commonly used to calibrate cloud lidar."*

***Changes***: *At lines 138 – 140 the following text was added: "While the backscatter coefficient includes scattering from both molecules and cloud particles, the OID is not sensitive enough to measure molecular only scattering (Ray and Anderson, 2015); therefore, it is assumed that all backscatter is from cloud particles."*

(144) "...the primary error source is likely the inversion of the range-resolved Lidar signal to estimate extinction." This is probably true and cast its shadow on the following. Suppose a Lidar Ratio (LR) of some tens sr, given the highest β values reported in the study, the extinction coefficient ε=LR*β (by the way, why use σ instead of ε which is more common in the literature?) would be larger than 10 km-1 and the attenuation from even a distance as short as 10 meters could be significant, and could explain some of the mismatch between computed and measured β, reported afterward. The authors should dwell more on how do they invert their lidar signal, what are the hypothesis done on the LR they use, what is their - at least qualitative - impact on the uncertainties. As instance, are they using the same LR for liquid and ice clouds? Unfortunately, the quoted reference Lolli et al (2013) is of no help since it deals with the determination of colour ratio of rain droplets, explicitly neglecting extinction effects.

*Attenuation does affect backscatter; however, since the OID uses a pulsed laser, close returns can be compared to returns from further away to assess if attenuation is a significant issue. This type of analysis indicates that attenuation affects the +7 case but not the other cases analyzed.*

***Changes:*** *The paragraph from line 133 to line 142 has been updated to make this information more clear.*

(259) "Backscatter efficiency values are calculated using MiePlot for diameters distributed log-normally between 1 μm and 30 mm." Not clear what "distributed lognormally" means here. Do you mean that the calculated efficiencies were calculated for radii equally spaced on a logarithmic scale from 1 μm to 30 mm?

*In this case, "distributed lognormally" is intended to mean that with progressively larger diameter particles, the intervals between diameters used in the calculation increases.*

**Changes:** *At lines 276 – 277 the following text was added: "Intervals within the smallest diameter channel (1 µm) are 0.0001 µm and increase to 3 µm intervals in the largest diameter channel (30 mm)."*

(260) "Backscatter efficiencies are averaged for all particle diameters within each channel" Where they arithmetically averaged? Was an attempt made to choose the mean value of the radius in the bin so that it was perhaps more representative? For example, by weighing the average of the radii with an estimate of the concentration of the particles at those radii, which can be derived for example from the estimated slope of the PSD in that bin (the arithmetic average assumes that the distribution of particles in the bin is unform). Could this make things better?

*The backscatter efficiencies are arithmetically averaged. As evident from Figure 3, the averaged backscatter coefficient efficiency changes very little from one bin channel to the next. Weighting the averaged efficiency by how the particles are distributed within the channel would move the efficiency slightly (we would estimate 10 %) to smaller sizes in the case of ice, where the efficiency mostly decreases with increasing size and hence may increase the overall backscatter. The maximum percentage difference between scattering efficiency changes from one channel to the next is 17%; hence, 10% of this would be 1.7%, which is small compared to the observed difference between ECP and OID backscattering.*

**Changes:** *Line 277 was updated to make explicit that the arithmetic mean is used for the backscatter efficiencies.*

(270) Figure 3 is quite interesting as it shows an increase of two orders of magnitude of the backscattering efficiencies for large particles, despite a relatively small change of the refractive index, from ice to water values. This was quite unexpected for me. I have taken the liberty of checking this result with one of the avatars of the BHMIE program which is at the core of the MiePlot package used in this work, and reproducing the same result. Still puzzled, I contacted Philip Laven (the author of the MiePlot package) who confirmed, with independent computation based on the Debye series approach, the correctness of the results of the paper. He explained that the 10th order rainbow is responsible for the increase in backscattering at 905 nm when the real part of the refractive index n = 1.3263 (the value chosen for water in the paper). The authors could underline the peculiarity of the factor 100 difference backscattered intensity at 905 nm between ice and water. In a sense, it is quite unfortunate that the choice of the 905 nm wavelength lead to such dramatic change in the backscattering from ice and liquid water, thus making the assumptions on the particulate phase very critical and impacting for the result. The reviewer thanks Philip Laven for the enlightening mail exchanges.

*The authors appreciate the extensive work which went into the verification of the results in Figure 3. The differences described were of some interest, and it is appreciated that there is an explanation. This have been further acknowledged in the text at lines*

**Changes:** *Lines 279 – 286 have been updated to further acknowledge the difference between water and ice backscatter efficiencies.*

(390) Figure 10 lower panel is not sufficiently addressed in the text. There it appear two regimes in the TWC-OID backscattering regression. The authors should dwell more on that, and perhaps define two different regression lines.

*The authors agree that the Fig. 10 should have been addressed further; particularly the lower panel. There does seem to be two different regimes in the OID backscatter coefficient regression: one for the cold data (primarily ice), and one for the warm data (primarily water). This would seem to be an important distinction as this is clearly not accounted for in the ECP backscatter coefficient shown in the top panel. Regression lines for water and ice have been added to the lower panel of Fig. 10, as well as discussion within the text.*

**Changes:** *To increase the readability of Fig. 10, the size of the scatter points has been reduced, and a level of transparency has been added to the markers on both the top and bottom of the figure. The x-axes of the top and bottom plots have been adjusted to match. The y-axes have been kept separate to maintain the one-to-one ratio of the top plot, as well as prevent a large amount of empty space for the bottom plot. A separate fit for the warm and cold data points has also been added in the bottom plot, with fits for the collective data removed.*

*Lines 402 – 412 and 523 - 534 have been updated to address the difference of performance in cold versus warm cases.*

**Responses to Anonymous Referee #2**

1. In line 244, what is "ni"? I think you refer to eta_i in equation (4). A similar issue appears in equation (5).

*This is a text error where "$n_i$" should be "$\eta_i$".*

**Changes:** *At line 254, "$n_i$" was changed to "$\eta_i$".*

2. For the upper right plot (B) in Figure 7, I think the median value is more reasonable than the mean value for the particle diameter measured by the ECP. Also, what is the definition of backscatter per second in Figure 5, the mean or median value?

*While the median will be different than the mean of the spectrum and could be viewed as a more reasonable way to represent the spectrum, the merged spectrum data currently do not contain the spectrum diameter median; hence, additional software development would be necessary. The point of Figure 7b is to show relative difference in spectrum diameters between the four cases and to show when/if size changes occur during the time period. For this purpose, the mean works as well as the median. Additionally, Figure 7c shows the particle spectrums so the reader can interpret how the mean and medium are different. Hence, we respectfully feel that it is not important for the reader's understanding to present the spectrum median instead of the spectrum mean in Figure 7b.*

*The backscatter per second for the OID is the result of 20 kHz measurements aggregated to 5 Hz raw data. The mean of the 5 Hz raw data is then taken to match the 1 Hz ECP data, noted in lines 142 – 143.*

***Changes:*** *No changes were made.*

3. The Figure 8 caption said the least square fit is the black line, but it is not black in Figure 8.

*This is an error where the "black" line should be labeled as "solid" (a change that was missed after the color scheme was updated to be more viewer friendly).*

***Changes:*** *The word "black" was changed to "solid" at line 416.*

4. Figure 10 needs some help. Two different color dots are heavily overlapped. Would it be possible to change to partly transparent to better distinguish cold and warm particles, or reduce the marker size? Furthermore, the figure with the same range for the x-axis and y-axis may be better to compare.

*The authors agree that several changes needed to be made to increase the readability of Fig. 10.*

***Changes:*** *To increase the readability of Fig. 10, the size of the scatter points has been reduced, and a level of transparency has been added to the markers on both the top and bottom of the figure. The x-axes of the top and bottom plots have been adjusted to match. The y-axes have been kept separate to maintain the one-to-one ratio of the top plot, as well as prevent a large amount of empty space for the bottom plot. A separate fit for the warm and cold data points has also been added in the bottom plot, with fits for the collective data removed.*

5. The study points out that the biased low calculated backscattering from ECP. The backscattering is calculated by the measured effective diameter in this study. I am not very clear how the effective diameter is determined in the measurement. You also mentioned a "fast-circle diameter method". However, I did not find the related description in the supplement materials either. I think it is better to describe the measure and convert process more.

*The diameter shown in the main text is found using the area-equivalent processing method, in which the total circular area of the pixels contained within an imaged particle are used to determine the associated diameter. The fast-circle processing method used in the supplemental material calculates the imaged particle diameter by encompassing the imaged particle entirely within a circle. The diameter of the resulting circle is assumed to be the diameter of the particle. Currently there is no generally agreed upon method for calculating the effective diameter, so we are presenting the two most accepted methods. Details regarding the methods for determining particle diameters will be added within the manuscript.*

***Changes:*** *Lines 217 – 222 have been updated to include the processing method details listed above.*

**Responses to Anonymous Referee #3**

**Major Comments:**

**Backscatter coefficient derivation from ECP data**

First of all, the assumption of ice sphere for the derivation of the backscatter coefficients from ECP data would cause a significant systematic bias. As confirmed by many aircraft observations of ice crystals, a majority of ice crystals have nonspherical shapes. The scattering properties of nonspherical ice crystals differ significantly from those of spherical ice. Although the authors claim that the uncertainty of the backscattering coefficients associated with the spherical ice assumption is much less than the uncertainty from particle size distribution (PSD) measurements (*Line 234–236*), these uncertainties would involve systematic biases, which will be carried over computing the backscattering coefficients from the PSD measurements.

To improve the validity of the present analysis for ice clouds, I suggest the authors add the following analysis to the ice cloud cases. To convert the extinction coefficients from backscattering coefficients through the inversion process, we use the lidar ratio ($S$; extinction-to backscatter ratio) of ice clouds that is empirically determined for each hydrometeour, and the latest value of the lidar ratio for ice clouds is, for example $S = 32$ sr at 532 nm (Holz et al., 2016). The lidar ratio at 905 nm could differ slightly from the one at 532 nm due to a slight difference of the real part of the ice refractive index between these wavelengths, but it should be quite consistent. The authors are strongly encouraged to perform the additional ECP data analysis with a lidar ratio of 32 sr for ice cloud cases.

Also, I noticed that the backscattering efficiency is introduced in Line 238 without a definition. Please clearly define the backscatter efficiency in the corresponding sentence. I believe that the authors defined the backscattering efficiency of a single particle as

$$Q_{back} = \frac{Q_{ext}\omega P_{11}(\pi)}{4\pi},$$ (R1)

where $Q_{ext}$ is the extinction efficiency; $\omega$ is the single-scattering albedo; and $P_{11}(\pi)$ is the scattering phase function at 180° degree. I would like to clarify if the authors include a denominator of $4\pi$. If the above is correct, the lidar ratio can be described as

$$S = \frac{Q_{ext}}{Q_{back}},$$ (R2)

Otherwise, I am wondering if the above definition differs from what is actually defined because the paper states that the fourth Stokes component $V$ is the focus of the study (Line 97).

*It is true irregular shaped ice crystals scatter differently than spherical ice crystals, and clouds have irregular shaped ice crystal as shown in the inserts of Fig. 4. There are two size parameters (radius/diameter) that are important: the radius in Eq. 4 of the paper for deriving backscatter coefficients and the diameter of the scattering efficiency at 180 degrees (Fig. 3). Typically these are the same; however, it is not apparent that they would be for non-spherical particles.*

*First, note that the uncertainties for determining particle diameter from 2-dimensional images are large (see the uncertainty analysis within the paper). The main manuscript uses an*

*area-equivalent diameter, and the supplemental material uses a fast-circle diameter. The area-equivalent diameter is typically more acceptable since it is the equivalent diameter of a sphere which has the same area as that observed within the 2-dimentational probe images. Hence, when converted to a radius, squared, and multiplied by π in Eq. 4, it provides the area of the imaged particle. There could be biases that are important to acknowledge in obtaining the area of a particle from a 2-dimentional image if the particles are not randomly oriented either in the probe images or in the Lidar backscatter volume. We have added an additional note on this potential bias in the manuscript.*

*There is also the diameter in the backscatter efficiency equation (Figure 3), which is currently the same as the diameter used in Eq. 4; however, since non-spherical and spherical particles do not scatter the same, a different diameter may be better to use as the equivalent backscatter diameter than the area equivalent diameter for obtaining the backscatter efficiency. Additionally, it is not clear that the backscatter coefficient in Eq. 4 should be related to the area equivalent diameter since the backscatter is dependent on surface waves, and the interference of the surface waves (see comment/discussion of Referee 1).*

*In conclusion, this manuscript uses the most accepted diameter for ice particles, and the supplement presents the fast-circle diameter for comparison, which we believe is the best that can be done. A note has been added to emphasize this within the text. While it is beyond the scope of this article, future work could use the ECP/OID data set to determine the diameter that provides the best agreement with the Lidar measurement for ice clouds.*

*The authors apologize as it was believed to be clear in the manuscript that a lidar ratio is not assumed for the calculations. The OID measures attenuated backscatter at different ranges and then fits a curve to the exponential function in Eq. 1. Since the lidar only samples along the wing of the aircraft (see Fig. 2), the cloud is homogenous enabling the curve fit. The attenuation is alpha, and the true, unattenuated backscatter is the y-intercept of the curve from the data fit. The lidar ratio is then derived from these two measured values. Therefore, the lidar ratio changes depending on the cloud conditions, and no single value is applied specifically for ice cases. Hence, the aircraft lidar processing is different than satellite based lidar processing where clouds sampled are not homogenous along the lidar beam at different range gates. We have included this information in the manuscript beginning at line 136.*

*This definition of the backscattering efficiency is correct, and a denominator of 4π is included. Eq. (5) has been added, along with corresponding text to indicate the normalization of the phase function.*

***Changes:** Line 304 -305 was edited to include "...or particles not being randomly oriented within the cloud probe measurement volumes."*

*Lines 443-447 have had additional text included to emphasis the main source of error: "The most likely systematic error relates to using Mie Theory and assuming cloud particles are spherical ice scatters. However, previous studies (Cairo et al., 2011) and theory (Mishchenko et al., 1996) would suggest that the ECP would be larger than the OID by a factor of two or more,*

*while the results indicate lower ECP backscatter coefficients. There are several other possible sources of error that could be produce a bias.".*

*Lines 218-222 have been added to emphasize the importance and difficulties in the method selected for the particle diameter calculations.*

*Lines 133 – 133 have been added to state the method used to derive the extinction coefficient.*

*Eq. (5) has been added at line 264.*

**Backscatter coefficient derivation from OID data**

As seen in Eq. (1) in the manuscript, the lidar signals from a certain location of ice clouds relative to the location of the aircraft can be attenuated by ice crystals in between the two locations. Therefore, the lidar signals need a correction with the two-way transmissivity to obtain the backscatter coefficient. The authors cite Lolli et al. (2013) for the extinction coefficient inversion for the present analysis, this paper is for rain droplets and the predefined lidar ratio for rain droplet (i.e., 50 sr in Lolli et al., 2013) may be inaccurate for small liquid droplets and ice clouds. Please add a few sentences describing how the extinction efficiency is derived for both liquid and ice cloud cases. In particular, what lidar ratios are used to estimate the extinction cross-section through the inversion of lidar measurements for each ice and liquid cloud case?

*The lidar ratio is derived by curve fitting the OID measured attenuation and the unattenuated backscatter, allowing for the lidar ratio to vary with the changing cloud conditions. As the sampling volume is relatively small, limited to only 10 m from the aircraft, attenuation by ice crystals is low. Thus, a correction with the two-way transmissivity is not necessary. The authors apologize that this is not clear in the text, and this information has been added.*

***Changes:*** *The above information has been added at lines 133 – 137.*

**Minor comments:**

1. "Hulst (1981)" should be "van de Hulst (1981)" throughout the manuscript.

   *This is an error by the authors and has been corrected*

   ***Changes:*** *"Hulst" has been changed to "van de Hulst at lines 95, 225, 253 and within the list of references.*

2. Lines 96–97 *"The 905 nm bean enables measurement of the fourth Stokes parameter (V) (Liou and Yang, 2016; Hulst, 1981) and is the focus of the study, ..."* I got an impression that the lidar instrument measures only the fourth component of the Stokes vector (*V*) from the manuscript. However, it actually measures the first component of the Stokes vector (*I*) in addition to the fourth component according to Ray and Anderson (2015), doesn't it? Please clarify it.

*It is correct that the OID measures both Stokes vector (I) and (V), however (I) was neglected to be mentioned as the results of (V) are the focus of this study. This has been clarified within the text.*

***Changes:*** *The text at lines 94 – 96 was updated to "With the 905 nm the OID is able to measure both the first Stokes parameter as well as the fourth Stokes parameter (Liou and Yang, 2016; van de Hulst, 1981). The fourth Stokes parameter is the focus of this study, and the 1550 nm wavelength channel is not used in the analysis."*

3. Lines 155-156 "*Images are produced when at least one array element is "shadowed" (i.e., reduced in intensity by 50% or more).*": Is there any reference that discusses the accuracy of estimated particle area from 2D-S with this approach?

*McFarquhar Et al. (2017) discusses this point on page 15, and Figure 11-7 using a CIP probe. Their results indicate that setting a threshold of 50 % results in derived particle diameters 100 µm less than those derived using a 70 % threshold, with this difference increasing with smaller sized particles. This citation has been added.*

***Changes:*** *The following text has been added at line 158: "This imaging threshold and it's accuracy are further discussed in McFarquhar et al. (2017)."*

4. Lines 187–190, Eq. (2): Use the italic font for scaler variables in the main text to be consistent with Eq. (2).

*This is a formatting error by the authors that has been corrected.*

***Changes:*** *All scalar variables throughout the manuscript have been italicized to match their respective equations.*

5. Line 226 *"geometric"* should be "geometric optics".

*This is an error by the authors and has been corrected.*

***Changes:*** *"geometric" has been changed to "geometric optics" at line 237.*

6. Line 234-236 *"While Mie theory …"*: Cairo et al. (2011) states in Page 561 that *"Generally speaking, aspherical scatterers depress the forward and back- ward scattering and enhance the side scattering with respect to surface equivalent spheres, so an overestimation of the backward scattering may be expected when using Mie codes. An educated guess of such overestimation can be provided by looking at studies comparing the phase function of aspherical vs spherical scatterers, which suggest an average overestimation of the Mie backscattering coefficient by a factor 2, which may possibly get as large as a factor 4 or more, depending on particle sizes and shapes (Mishchenko et al., 1996)."* Please revise the corresponding sentence to be consistent with the statement by Cairo et al. (2011).

*To be more consistent with Cairo et al. (2011) and Mischenko et al. (1996), the text has been updated.*

*Changes:  Lines 241 – 245 have been updated for consistency.*

7. Lines 246-247 "an equivalent sphere": This should be clearly stated as "a projected area equivalent sphere" in order to avoid confusion with a volume equivalent sphere. The backscattering coefficient is proportional to the cross-sectional area of a particle for large size parameters (i.e., *Qext.* = ~2), so that the use of projected area equivalent spherical radius is relevant for both liquid and ice particles in the present analysis.

*The authors respectfully disagree that this should be considered as a projected area equivalent sphere, as the surface waves mentioned in line 265 prevent $Q_{back}$ from reaching an asymptote and becoming proportional to the cross-sectional area of the particle. Instead, a more appropriate labeling in line 246 would be "a backscatter equivalent sphere".*

*Changes: Line 256 has been updated from "…an equivalent sphere…" to "…a backscatter equivalent sphere…".*

8. Lines 267–268: This statement is inconsistent with Lolli et al. (2013) that use a predefined extinction-to-backscatter ratio (or the lidar ratio) to estimate the extinction coefficients from lidar signals. Thus, the backscattering efficiency is necessary for to interpret OID data (as clearly indicated in Eq. 1).

*In Eq. 1, an assumed lidar ratio is not necessary. The OID measures attenuated backscatter at various ranges and then fits this curve to the exponential function in Eq 1. The attenuation is alpha, and the true unattenuated backscatter is the y-intercept of the curve. The lidar ratio is then derived from these two measured values. Thus, while the backscatter efficiency is necessary to calculate the ECP backscatter coefficients, it is not necessary for understanding the OID data. However, the in question is not necessary in the text, and does not make this point clearer. To prevent ambiguity, those lines have been removed.*

*Changes: The statement at lines 275-277 has been removed, and further information on the calculation of the OID extinction coefficient has been added to lines 133 – 137.*

9. Figure 3 caption: 0.0001 µm should be 0.0001 mm. Also, 3 µm should be 3 mm.

*The values 0.0001 µm and 3 µm are intentional as they are to indicate the intervals between diameters used for the individual backscatter efficiency calculations, not the intervals in diameters between the average backscatter efficiency calculations. This was not clear in the text, thus both the text and the Fig. 3 caption have been updated.*

*Changes: At lines 276 – 277 the following text was added: "Intervals within the smallest diameter channel (1 µm) are 0.0001 µm and increase to 3 µm intervals in the largest diameter channel (30 mm)." Lines 291 – 292 have been updated to "Intervals between diameters used for the individual backscatter efficiency calculations range from 0.0001 µm at the lowest diameters, to 3 µm at the highest diameters."*

10. Figure 5a: What lidar ratio is used to derive the extinction coefficient for the OID analysis for this case? As Lolli et al., (2013) use a lidar ratio of 50 sr for rain drop that is

significantly larger than those of cloud liquid droplets (~20 sr), the two-way transmissivity could be overestimated, so that OID derived backscattering coefficient might be underestimated. The authors are encouraged to clarify this.

*As mentioned previously in comment 8, a lidar ratio is not assumed but rather repeatedly derived by using the measured attenuation and the true unattenuated backscatter in a curve fit. Thus, the lidar ratio changes to match the environment which encounters little attenuation due to the relatively small sampling volume.*

**Changes:** *Further information on the calculation of the OID extinction coefficient has been added to lines 133 – 137.*

11. Line 378 *"..., which indicates an unaccounted source of systematic error."*: I think this may be due to the backscattering efficiency bias associated with an ice sphere assumption. In Lines 295-296 the manuscript says *"Eq (7) does not include systematic errors (e.g., uncertainty in backscatter efficiency)"*. I suggest the authors to mention that one of unaccounted errors would be a systematic bias in backscatter efficiency.

*While a backscattering efficiency bias likely has an effect to some degree, as mentioned in comment 6 and in Cairo et al. (2011), calculated backscatter coefficients using Mie theory code on aspherical particles should be a factor of at least 2 higher than those that are measured. Thus, the backscatter efficiency bias cannot fully explain why the OID backscatter coefficients are higher than those calculated from ECP data. However, this bias is still important to acknowledge.*

**Changes:** *Lines 314 – 316 have been updated to "The third standard deviation is used as a threshold in accordance with the Three Sigma Rule (Pukelsheim, 1994) since the OID is in development and Eq. (7) does not include systematic errors (e.g. systematic due to spherical particle assumptions)."*

**References**

Holz, R. E., and Coauthors, 2016. Resolving ice cloud optical thickness biases between CALIOP and MODIS using infrared retrievals. Atmospheric Chemistry and Physics, 16(8), 5075-5090.

McFarquhar, G. M., and Coauthors, 2017: Processing of Ice Cloud In Situ Data Collected by Bulk Water, Scattering, and Imaging Probes: Fundamentals, Uncertainties, and Efforts toward Consistency. *Meteorological Monographs*, **58**, 11.1-11.33, https://doi.org/10.1175/AMSMONOGRAPHS-D-16-0007.1.

---

## Referee Report (RR1)

Manuscript number: amt-2022-087
Full title: Technique for comparison of backscatter coefficients derived from in-situ cloud prove measurements with concurrent airborne Lidar
Author(s): Wagner and Delene

The authors have provided adequate explanations to most of my comments. However, there are still a couple of unclear descriptions in the current manuscript, which should be improved before publication. The topic presented in this paper is suitable for Atmospheric Measurement Techniques. I recommend Minor Revisions for publication.

**Comments**
*Backscatter coefficient derivation from ECP data*

1. The authors' response to my first comment on the backscattering coefficient derivation from ECP data went off what was supposed to be. A main focus of the first comment was the definition of the backscattering efficiency. Although I suggested the authors to clearly define the backscatter efficiency in the manuscript in the previous round of review, it has not been specified in the current manuscript. The below definition of the backscattering efficiency is commonly used for remote sensing of ice clouds based on micro pulse lidar observations

$$Q_{back} = \frac{Q_{ext}\omega P_{11}(\pi)}{4\pi}, \qquad (R1)$$

where $Q_{ext}$ is the extinction efficiency; $\omega$ is the single-scattering albedo; and $P_{11}(\pi)$ is the scattering phase function at 180° degree. Substituting Eq (R1) into $Q_i$ in Eq (4) gives exactly the backscattering coefficient under the assumption of the projected-area-equivalent sphere radius in Eq. (4), as the extinction/scattering/absorption/backscattering efficiencies are the quantities relative to the projected area of a particle. Therefore, I agree with the authors' statement that the area-equivalent sphere diameter/radius is typically more acceptable. Also, I would like to argue that the backscattering coefficient should be related to the projected area of a particle (i.e., should be the area-equivalent radius in Eq. 4).

    In addition, 180 in Eq (5) should be $2\pi$ due to the radian unit in trigonometric functions. Please improve the corresponding descriptions.

    On the backscattering coefficient derivation from OID data, I read Ray and Anderson (2015) and understand that the lidar ratio is derived by curve fitting of the two-way attenuated backscattering intensity measured from OID. Although the authors' response to the comment were somewhat inconsistent with what the paper described, the corresponding descriptions in the revised manuscript are now all consistent and clear.

2. I am confused with an inconsistent description in the revised manuscript that *"For water spheres, $\pi r^2$ is the cross-sectional area (A), while for irregular particles such as ice, A is modeled as the cross-sectional area of a backscatter equivalent sphere."* What is a backscatter equivalent sphere? The quantity $A$ must be a geometric cross-sectional area of a particle regardless of their particle shapes, as both liquid and ice cases rely on Eq. (4) in deriving the backscattering coefficients from ECP data. As this is critical, please clarify and improve the inconsistency.

3. Figure 3 caption: "**A refractive index of 1.3263 + 5.6 x 10$^{-7}$j**" To express the imaginary quantity, $i$ should be used instead of $j$.

---

## Author Response (AR3)

**Amt-2022-87 Authors' Responses to Anonymous Referee #3**

The authors would like to thank anonymous referee #3 for continuing to provide thoughtful feedback on the manuscript contents. The original comments by referee #3 are presented first, followed by the author response in italic font and then the respective changes to the manuscript.

**Responses to Anonymous Referee #3**

The authors have provided adequate explanations to most of my comments. However, there are still a couple of unclear descriptions in the current manuscript, which should be improved before publication. The topic presented in this paper is suitable for Atmospheric Measurement Techniques. I recommend Minor Revisions for publication.

**Comments**
*Backscatter coefficient derivation from ECP data*
1. The authors' response to my first comment on the backscattering coefficient derivation from ECP data went off what was supposed to be. A main focus of the first comment was the definition of the backscattering efficiency. Although I suggested the authors to clearly define the backscatter efficiency in the manuscript in the previous round of review, it has not been specified in the current manuscript. The below definition of the backscattering efficiency is commonly used for remote sensing of ice clouds based on micro pulse lidar observations

$$Q_{back} = \frac{Q_{ext}\,\omega\,P_{11}(\pi)}{4\pi}, \qquad\qquad (R1)$$

where $Q_{ext}$ is the extinction efficiency; $\omega$ is the single-scattering albedo; and $P_{11}(\pi)$ is the scattering phase function at 180° degree. Substituting Eq (R1) into $Q_i$ in Eq (4) gives exactly the backscattering coefficient under the assumption of the projected-area-equivalent sphere radius in Eq. (4), as the extinction/scattering/absorption/backscattering efficiencies are the quantities relative to the projected area of a particle. Therefore, I agree with the authors' statement that the area-equivalent sphere diameter/radius is typically more acceptable. Also, I would like to argue that the backscattering coefficient should be related to the projected area of a particle (i.e., should be the area-equivalent radius in Eq. 4).

In addition, 180 in Eq (5) should be $2\pi$ due to the radian unit in trigonometric functions. Please improve the corresponding descriptions.

On the backscattering coefficient derivation from OID data, I read Ray and Anderson (2015) and understand that the lidar ratio is derived by curve fitting of the two-way attenuated backscattering intensity measured from OID. Although the authors' response to the comment were somewhat inconsistent with what the paper described, the corresponding descriptions in the revised manuscript are now all consistent and clear.

*The authors feel that they now understand what is being requested of the backscatter efficiency equation, which has been added to the manuscript. Updates have also been made to further clarify that Eq. 4 is utilizes the area-equivalent radius. It is correct that Eq. 5 should integrated to $2\pi$ and this has been updated in the manuscript.*

***Changes:*** *Line 254 has been updated to specify that $r_i$ is the area-equivalent particle radius. The backscatter efficiency equation has been added as Eq. 5, with all following equations adjusted up*

*one number and a description of Eq. 5 added at lines 260 - 264.  Eq. 6 has been updated to replace "180" with $2\pi$.*

2. I am confused with an inconsistent description in the revised manuscript that *"For water spheres, $\pi r2$ is the cross-sectional area (A), while for irregular particles such as ice, A is modeled as the cross-sectional area of a backscatter equivalent sphere."* What is a backscatter equivalent sphere? The quantity *A* must be a geometric cross-sectional area of a particle regardless of their particle shapes, as both liquid and ice cases rely on Eq. (4) in deriving the backscattering coefficients from ECP data. As this is critical, please clarify and improve the inconsistency.

*The authors mean to indicate that an area-equivalent sphere is used to represent the size of the ice particle since Mie theory applies to spheres. To keep the text consistent the text has been updated with further details on how the cross-sectional area and diameter of an ice crystal are determined.*

***Changes:*** *Lines 255 – 258 have been updated to "For water spheres, $\pi r^2$ is the cross-sectional area (A), while for irregular particles such as ice, A is the cross-sectional area imaged by the two-dimensional probes (see the ice crystal image insert in Fig. 1). The cross-sectional area is converted to a diameter (or radius) by determining the sphere that has an equivalent, two-dimensional projected surface area to A.*

3. Figure 3 caption: "**A refractive index of 1.3263 + 5.6 x 10-7j**" To express the imaginary quantity, *i* should be used instead of *j*.

*This is correct, and the manuscript has been updated to use "i" to represent the imaginary quantity.*

***Changes:*** *Lines 265, 266, 297 and 298 have been updated to replace "1.3263 + 5.6 x 10-7j" with "1.3263 + 5.6 x 10-7i" and "1.3031 + 5.6 x $10^{-7}j$"  "1.3031 + 5.6 x $10^{-7}i$".*